



# Storm track response to uniform global warming downstream of an idealized sea surface temperature front

Sebastian Schemm[1], Lukas Papritz[1], and Gwendal Rivière[2]

[1]Institute for Atmospheric and Climate Science, ETH Zürich, Zurich, Switzerland
[2]LMD/IPSL,École Normale Supérieure, PSL Research University, Sorbonne Université, École Polytechnique, CNRS, Paris, France

**Correspondence:** Sebasian Schemm (sebastian.schemm@env.ethz.ch)

**Abstract.** The future evolution of storm tracks, their intensity, shape, and location, is an important driver of regional precipitation changes, cyclone-associated weather extremes, and regional climate patterns. For the North Atlantic storm track, Coupled Model Intercomparison Project (CMIP) data indicate a tripole pattern of change under the RCP 8.5 scenario. In this study, the tripole pattern is reproduced by simulating the change of a storm track generated downstream of an idealized sea surface temperature (SST) front under uniform warming on an aquaplanet. The simulated tripole pattern consists of reduced eddy kinetic energy (EKE) upstream and equatorward of the SST front, extended and poleward shifted enhanced EKE downstream of the SST front, and a regionally reduced EKE increase at polar latitudes. In the absence of the idealized SST front, in contrast, the storm track exhibits a poleward shift but no tripole pattern. A detailed analysis of the EKE and eddy available potential energy (EAPE) sources and sinks reveals that the changes are locally driven by changes in baroclinic conversion rather than diabatic processes. However, globally the change in baroclinic conversion averages to zero, thus the observed global EAPE increase results from diabatic generation. In particular, resolved-scale condensation plus parameterized cloud physics dominate the global EAPE increase followed by longwave radiation. Feature-based tracking provides further insight into cyclone life-cycle changes downstream of the SST front. Moderately deepening cyclones deepen less in a warmer climate, while strongly deepening cyclones deepen more. Similarly, the average cyclone becomes less intense in a warmer climate, while the extremely intense cyclones become more intense. Both results hold true for cyclones with genesis in the vicinity of the SST front and elsewhere. The mean cyclone lifetime decreases, while it increases for those cyclones downstream of the SST front. The mean poleward displacement between genesis and maximum intensity increases for the most intense cyclones, while averaged over all cyclones there is a mild reduction and the result depends on the definition of the displacement. Finally, the number of cyclones decreases by approximately 15 %. Aquaplanet simulations with a localized SST front thus provide an enriched picture of storm track dynamics and associated changes with warming.

## 1 Introduction

The variability of storm tracks is of paramount importance in understanding regional climate patterns and their change (Shepherd, 2014). A complete mechanistic understanding of projected future storm track changes is still lacking because main factors underlying the variability of storm tracks compete with each other (Shaw et al., 2016; Shaw, 2019). Scientific attention has thus





shifted to the analysis of these forcing factors across model hierarchies and to studies that attempt to isolate the importance of
individual factors for future storm track activity.

Individual factors contributing to projected storm track changes have received enhanced consideration in recent years in
particular lower and upper-level baroclinicity and increased diabatic processes. Enhanced upper-level baroclinicity, which
results from lower stratospheric cooling and tropical warming, is found to make waves with low wavenumbers more unstable
and short wavenumbers more stable with the former being more capable of pushing the jet poleward (Rivière, 2011). Thus, a
poleward shift and a change in the eddy length scale is observed under warming (Kidston et al., 2010; Rivière, 2011). This was
recently confirmed by Chemke and Ming (2020), noting that large-scale waves will be stronger at the end of the century, while
the opposite was found for small-scale waves. However, an increase in humidity favors the development of more cyclonically
breaking waves, which thus may – at least partly – counteract the previous mechanism as these waves push the jet equatorward
(Orlanski, 2003). Further, there is good reason to assume that the tropical widening in response to global warming, which is
a robust feature in climate models (Lu et al., 2007; Hu et al., 2013), adds to the poleward displacement of the midlatitude
storm tracks (Kushner et al., 2001; Yin, 2005; Simpson et al., 2014). Indeed, increased dry static stability in the tropics is
able to shift the storm tracks poleward, which may or may not occur in tandem with a widening of the Hadley cell (Mbengue
and Schneider, 2013). Similar holds true for increased subtropical stability on the equatorward side of the jet in general (Lu
et al., 2007, 2010). More recently, the enhanced poleward propagation of extratropical cyclones has been pinpointed as another
crucial factor for the poleward shift of the storm tracks (Tamarin-Brodsky and Kaspi, 2017). This is motivated by the fact
that increased deepening rates are inherently connected to enhanced poleward motion (Gilet et al., 2009; Coronel et al., 2015;
Tamarin and Kaspi, 2016; Besson et al., 2021). Consequently, simulations with idealized warming or positive SST anomalies,
which are both key ingredients known to intensify cyclone deepening rates, all display an enhanced poleward deflection and
shift of the storm tracks, at least for the very intense storms (Palmer and Zhaobo, 1985; Brayshaw et al., 2008; Kodama and
Iwasaki, 2009; Graff and LaCasce, 2012; Tamarin and Kaspi, 2017).

The two main storm tracks in the Northern Hemisphere are located over the North Atlantic and North Pacific Oceans and
downstream of the western boundary currents, i.e., the Gulf Stream and Kuroshio Extension. The western boundary currents
drive and anchor the oceanic storm tracks as they maintain a near-surface zone of enhanced baroclinicity and supply heat and
moisture from below by means of sensible and latent heat fluxes (Chang et al., 2002; Sampe et al., 2010; Brayshaw et al., 2011;
Papritz and Spengler, 2015). The SST fronts locally reinforce the meridional temperature gradient by warming via sensible
heat flux where it is warmer and cooling where it is colder (Hotta and Nakamura, 2011). Globally, the meridional temperature
gradient is maintained by the differential solar irradiance. Locally, the combination of enhanced moisture storage capacity in
a warmer climate and latent heat supply from below makes SST fronts prone to explosive moist cyclogenesis, in particular
the downstream sector is of interest because condensation in extratropical cyclones occurs later during the life cycle. However,
while there is on one side evidence that diabatic heating within storms tends to intensify the growth of transient eddies (e.g., Kuo
et al., 1991; Davis et al., 1993; Stoelinga, 1996; Chang et al., 2002; Schemm et al., 2013) transient diabatic processes including
surface flux have at the same time been found to reduce transient eddy available potential (EAPE) and thereby reduce the
reservoir from which eddy kinetic energy (EKE) can be generated (Ulbrich and Speth, 1991; Chang and Zurita-Gotor, 2007;





Marcheggiani and Ambaum, 2020). This might occur, for example, if diabatic heating occurs on the poleward side of the jet and thereby reduce baroclinicity (i.e., the mean available potential energy) (Laîné et al., 2011). This influence of the transient diabatic heating by moist processes must not be confused with the diabatic influence of the hemispheric-wide radiation that make a positive contribution to the mean available potential energy (Oort, 1964; Oort and Peixóto, 1974). Hence, on the scale of individual storms, increased diabatic heating in a warmer climate must also occur "in the right location" to help with the

deepening (Kirshbaum et al., 2018). Further, individual storms must optimize their vertical tilt magnitude and tilt orientation to maximize their baroclinic growth efficiency (Davies and Bishop, 1994; Schemm and Rivière, 2019). The baroclinic conversion efficiency relates to the eddy diffusivity of storm tracks (Eq. 9 in Schemm and Rivière, 2019), which is a key quantity that controls the amplitude and transport characteristics of baroclinic eddies (Held and Larichev, 1996; Lapeyre and Held, 2003; Schemm and Schneider, 2018). Its response to warming is largely unknown, but changes in the vertical structure of storms have

been reported by earlier idealized studies (Tierney et al., 2018; Kirshbaum et al., 2018; Sinclair et al., 2020) suggesting that the baroclinic conversion efficiency and thus the storm-track diffusivity might change under warming (Caballero and Hanley, 2012).

The storm track response downstream of western boundary currents is complex and might differ from the storm track response seen in zonally uniform idealized warming experiments. The extent to which the storm tracks that form downstream

of a major SST front react differently than others remains open though some studies already give an indication. For example, Brayshaw et al. (2008) and Graff and LaCasce (2012) showed that an increase in the SST gradient in the extratropics increases the strength of the storm tracks but only if the SST gradients increase not too close to the subtropical jet. Uniform idealized global warming experiments, which do not change the SST gradients, have yielded mixed results. In the study of Graff and LaCasce (2012) and Sinclair et al. (2020), the storm tracks appear to shift poleward and strengthen and extend vertically. In

Kodama and Iwasaki (2009) the jet and the storm track also shift poleward and the upper-level meridional temperature gradient increases due to enhanced tropical heating (in agreement with Sinclair et al., 2020), but the global mean eddy activity does not change in Kodama and Iwasaki (2009).

**Main research questions**

In this study, a localized SST gradient is added to an otherwise zonally uniform SST distribution to mimic a western boundary

current and to create a regional storm track. This study aims to establish the archetypal response of a storm track generated downstream of a SST front to uniform global warming. The experiment is motivated by the fact that the two main storm tracks in the Northern Hemisphere originate downstream of western boundary currents. The reduced complexity of an aquaplanet appears attractive to provide a clearer picture of the partially opposing process-level changes found in studies based on CMIP data. Future studies will explore the sensitivity of the result to the SST front gradient, strength and position but here the

focus is on the response to simple uniform warming. The response is described in terms of the well-known eddy energy conversion cycle, including baroclinic, barotropic and diabatic conversion into EAPE and into EKE by the former two and also the baroclinic conversion efficiency. The eddy energy cycle is widely used and allows for a comparison with existing studies. Further, we analyse basic aspects from a feature-based perspective, such as changes in the life-time, deepening rates and





poleward displacement. We systematically compare cyclones that develop downstream of the SST front to all other cyclones,
in particular to those in the opposite hemisphere, for which we do not prescribe a SST front.

## 2    Simulation design and diagnostics

### 2.1    Idealized aquaplanet simulations with SST front

This study uses the Icosahedral Nonhydrostatic Weather and Climate Model (ICON; Zängl et al., 2015) initially developed in a
joint collaboration between the German Weather Service (DWD) and the Max Planck Institute for Meteorology (MPI-M) and
also in pre-operational use at the Swiss National Weather Service (MeteoSwiss). The model runs on an icosahedral grid with an
effective horizontal resolution of approximately 80 km (called R02B05 in the ICON framework). It has 70 levels in the vertical
and the model top is placed at 65 km. The model employs a set of standard sub-grid scale parameterizations used for numerical
weather prediction purposes. These include in particular a one-moment two-category microphysics scheme (Doms et al., 2011),
a convection scheme following Tiedtke (1989), a prognostic TKE scheme for sub-gridscale turbulent transfer (Raschendorfer,
2001), and a parameterization for non-orographic gravity wave drag (Orr et al., 2010). For radiation, the ecRad scheme is used
(Hogand and Bozzo, 2018).

    The aquaplanet's atmosphere is initialized with a standard configuration set: The atmosphere following the equations given
by the Jablonowski-Williamson baroclinic wave test case (Jablonowski and Williamson, 2006) and the background SST fol-
lowing the well-known "Qobs" distribution by Neale and Hoskins (2001), which approximates the observed zonal mean SST
distribution. In the Northern Hemisphere, an idealized SST front is superposed on the "Qobs" distribution. This SST front
consists of two ellipsoids of the same size but reversed sign. The maximum amplitude at the centers of the ellipsoids is 10 K
decaying exponentially with distance. The mid-point between the two ellipsoids is placed at $30°$W and $42°$N and their centers
are shifted against each other by $7°$ latitude. In order to mimic the behaviour of the Gulf Stream front, the ellipsoids are rotated
by $25°$ wrt. a longitude circle. Finally, the minimum temperature is limited to the freezing temperature of 273.15 K. No SST
front is placed in the Southern Hemisphere to allow for a direct comparison with a front-free zonally symmetric atmosphere.
The zonally symmetric background SST profile, the SST front and the 2-m temperature anomaly computed as deviation from
the zonal mean of the climatological 2-m temperature are shown in Fig. A1. The 2-m temperature anomaly will be shown in
each figure throughout the manuscript (see for example solid and dashed contours in Fig. 1).

    Two simulations are performed: One following the setup as just described and one with a uniform increase of the SST of
4 K. For each simulation, the model is spun-up for a month and then a total of 20 winter seasons is simulated, corresponding
to 60 simulated perpetual months. Note that the there is no seasonality in the model and the solar zenith angle at the equator at
noon is kept at constant $90°$.





## 2.2 Eddy kinetic and eddy available potential energy budget

To understand the change in eddy kinetic (EKE) and eddy available potential energy (EAPE), tendencies for EKE and EAPE

are computed using the equations already used in Chang et al. (2002); Drouard et al. (2015); Rivière et al. (2018); Schemm and Schneider (2018); Schemm and Rivière (2019), which go back to the work by Cai and Mak (1990) and Orlanski and Katzfey (1991).

### 2.2.1 Eddy kinetic energy equation

The EKE and EAPE budget have been derived before and are presented here for completeness. The EKE budget equation

is obtained by highpass-filtering of the primitive equations of motion in pressure coordinates and after multiplication by the highpass-filtered horizontal wind components. A 10-day frequency cut off is used in the highpass filter. The resulting tendency equation, where primes denote highpass-filtered and overbars lowpass-filtered quantities, is,

$$\frac{\partial K_e^{'}}{\partial t} = -\nabla \cdot (\mathbf{v} K_e^{'} + \mathbf{v}_a^{'} \phi^{'}) - \frac{\partial}{\partial p}(\omega K_e^{'} + \omega^{'} \phi^{'}) + \omega^{'} \frac{\partial \phi^{'}}{\partial p} - \mathbf{v}^{'} \cdot (\mathbf{v}_3^{'} \cdot \nabla \overline{\mathbf{v}}) + R_{K_e}, \tag{1}$$

where the residual term is $R_{K_e} = \mathbf{v}^{'} \cdot (\overline{\mathbf{v}_3 \cdot \nabla \mathbf{v}} - \overline{\mathbf{v}}_3 \cdot \nabla_3 \overline{\mathbf{v}})$, $\phi^{'}$ the filtered geopotential height. Three dimensional vectors have

a subscript "3". The sum of the three terms $-\frac{\partial}{\partial p}(\omega^{'} \phi^{'}) + \omega^{'} \frac{\partial \phi^{'}}{\partial p} - \nabla(\mathbf{v}_a^{'} \phi^{'})$, where $\mathbf{v}_a^{'}$ is the ageostrophic wind, denotes the pressure gradient force $(-\mathbf{v}^{'} \cdot \nabla \phi^{'})$. The first term on the r.h.s denotes the flux horizontal convergence of EKE, the second the horizontal ageostrophic geopotential flux and the third and fourth term the corresponding vertical fluxes. The fifth term is the internal baroclinic conversion from EAPE to EKE and the sixth term the barotropic conversion. The last term, finally, is the residual accounting for numerical errors in the computation of the EKE budget. If the time mean is used instead of a highpass

filter, the equation reduces to the EKE tendency of Orlanski and Katzfey (1991). More details on the derivation are presented in the Appendix of Schemm and Rivière (2019).

### 2.2.2 Eddy available potential energy equation

Similarly an equation for the EAPE tendency is obtained by multiplying the highpass-filtered thermodynamic equation with $\theta^{'}/S$ (Chang et al., 2002; Drouard et al., 2015), which yields

$$\frac{\partial P_e^{'}}{\partial t} = -\nabla \cdot (\mathbf{v} P_e^{'}) - \omega^{'} \frac{\partial \phi^{'}}{\partial p} - \frac{1}{S} \theta^{'}(\mathbf{v}^{'} \cdot \nabla \overline{\theta}) - \frac{1}{S} \frac{\partial}{\partial p}\left(\omega \frac{\theta^{'2}}{2}\right) + \frac{1}{S} \theta^{'} Q^{'} + R_{P_e}, \tag{2}$$

where the residual term is $R_{P_e} = \frac{\theta^{'}}{S}[\overline{(\mathbf{v}_3 \cdot \nabla \theta)} - \overline{\mathbf{v}}_3 \cdot \nabla_3 \overline{\theta}]$. The diabatic source term is denoted by $Q^{'}$ and includes short- and longwave radiation $Q_{RD}^{'}$, parameterizations of convection $Q_{CN}^{'}$, turbulence $Q_{TR}^{'}$, microphysics $Q_{MP}^{'}$ and a resolved-scale saturation adjustment scheme that produces condensation and evaporation $Q_{SA}^{'}$. The saturation adjustment is called before and after the dynamical core in the model. Their contributions to the build-up of EAPE will be discussed in detail in this

study. The tendencies from gravity wave drag, which are small, are neglected and the tendencies from subgird-scale orography are not existing on an aquaplanet. The static stability parameter is defined as $S = -h^{-1}\frac{\partial \theta_R}{\partial p}$. The definition makes partly use of quasi-geostrophic scaling assumptions and of a vertical reference potential temperature computed from the climatological





mean of the simulation. The scale height is $h = \frac{R}{p}(\frac{p}{p_0})^{R/c_p}$. The second term on the r.h.s is the internal baroclinic conversion from EAPE into EKE - the same as in Eq. 2.2.1 but with opposite sign - , the third term is the external baroclinic conversion

representing the conversion of mean available potential energy into EAPE, and the fourth term is the vertical flux convergence. The last term again denotes the residual. The analysis in the following sections puts its emphasis on the conversion terms rather than on advection and ageostrophic geopotential flux. The later only redistribute energy and are not global sources and sinks.

The external baroclinic conversion into EAPE is the scalar product between the eddy heat flux and the mean baroclinicity. It can be decomposed into contributions from the mean baroclinicity, the eddy total energy $T_e^{'}$, which is given by the sum of

EAPE and EKE, and the baroclinic conversion efficiency (Rivière et al., 2018),

$$\frac{1}{S}\theta^{'}(\mathbf{v}^{'} \cdot \nabla \overline{\theta}) = T_e^{'}|\mathbf{B}_s|E_{ff},\tag{3}$$

where the eddy efficiency $E_{ff}$ is defined as

$$E_{ff} = \frac{|\mathbf{Q}_s|}{T_e^{'}}\cos(\mathbf{Q}_s, \mathbf{B}_s),\tag{4}$$

where $\mathbf{Q}_s = \frac{1}{\sqrt{S}}(\mathbf{v}^{'}\theta^{'})$ is the eddy heat flux normalized by the static stability and $\mathbf{B}_s = -\frac{1}{\sqrt{S}}\nabla \overline{\theta}$ is the background baroclin-

icity. The eddy efficiency is independent of the eddy amplitude and thus, in contrast to the eddy heat flux, particularly useful for the study of baroclinic conversion changes. The efficiency is maximized if the magnitude and orientation of the vertical westward tilt with height is such that the resulting eddy heat flux ($\mathbf{Q}_s$) aligns with the mean baroclinicity $\mathbf{B}_s$ (see Fig. 1 in Schemm and Rivière, 2019). Notably, the time mean of the eddy efficiency relates to the eddy diffusivity in storm tracks (Eq. 9 in Schemm and Rivière, 2019), which is known to control the amplitude and meridional transport characteristics of baroclinic

eddies (Held and Larichev, 1996; Lapeyre and Held, 2003; Schemm and Schneider, 2018)

## 2.3   Feature-based cyclone tracking

The feature-based method to detect surface cyclones is based on a contour search approach as introduced by Wernli and Schwierz (2006). The scheme identifies local minima in the sea level pressure field (SLP), which are enclosed by a closed SLP contour. Subsequently, these minima are tracked in time at 6-hourly temporal resolution. As in previous studies, the minimum

lifetime of a track is one day. Further refinements related to splitting and merging of tracks that extend the original scheme are described in Sprenger et al. (2017). These refinements do not affect the results of this study. The total number of analyzed cyclone tracks from the 20 simulated winter seasons is about 15,900 in the control simulation and 13,500 in the warmed simulation.

The identified surface tracks are used to study deepening rates, which are defined as the change of SLP along a track

computed between two consecutive 6-hourly times steps. Additionally considered are maximum intensities, defined as the minimum SLP value found along a track, and the life time and meridional displacements. The latter is defined once as the difference in latitude between genesis and lysis (corresponding to the first and last time step along a track, respectively) and the difference between the genesis latitude and the latitude of maximum intensity.



## 3 Results

### 3.1 Storm track response to warming on aquaplanet compared with CMIP5 data

The SST front on the aquaplanet anchors a locally enhanced storm track that extends from slightly upstream of the center of the front near 30°E downstream toward 90°E (Figs. 1a and b). Further downstream near 150°E, 250-hPa EKE values begin to decrease and approach those found in the undisturbed Southern Hemisphere. EKE begins to strengthen again near 150°W. Near 30°N / 30°E there is an indication of the formation of a stationary trough, as predicted by theory, because the midlatitude atmosphere tends to offset low-level warming by horizontal advection of cold air from the north, which requires the presence of a low pressure anomaly located downstream of the heat source (Hoskins and Karoly, 1981). Aside from this slight trough anomaly, the storm track appears zonally symmetric, as would be expected on an aquaplanet. We note that the overall patterns are similar also on different levels, such as the 500 hPa level but with a reduced magnitude. Overall, the simulated interactions between the idealized SST front and the extratropical circulation result in a local intensification of the EKE that affects the entire depth of the troposphere and extends far downstream of the SST front. The response of this storm track downstream of the SST front to idealized global warming is the focus of the following analysis.

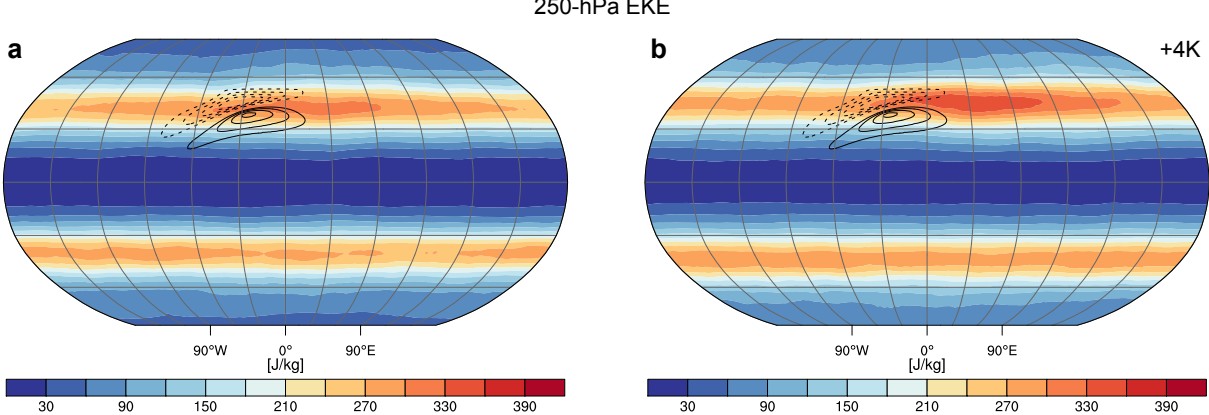

**Figure 1.** Eddy kinetic energy (shading; J kg$^{-1}$) at the 250 hPa level for the (a) Qobs control run with SST front in the Northern Hemisphere and (b) with homogeneous warming by 4 K at the surface. The SST front is indicated in this and the following figures by its imprint in the 2-m temperature after removing the zonal mean (black solid contours indicate positive values and dashed negative values; starting at ±0.5 K in steps of 0.5 K).

The 250-hPa EKE change between the control scenario and the warming scenario, consisting of a simple uniform surface warming of 4 K, shows three distinct anomaly patterns that are also found in fully coupled CMIP5 simulations under the RCP8.5 greenhouse gas emission scenario. The EKE change at the end of the century under the RCP8.5 scenario compared to present-day conditions (Fig. 2) shows a poleward shift and intensification of the storm track in the Southern Hemisphere, which has been documented in previous literature (Chang et al., 2012). While the magnitude of the EKE change differs among





the models selected here, the pattern does not qualitatively change when all models and all ensemble members are included (see Fig. 1 in Tamarin and Kaspi (2017)), the overall pattern of EKE changes exhibits similarities. For example, EKE decreases over the North Atlantic equatorward of the Gulf Stream SST front (labeled "1" in Fig. 2), while it increases downstream and northeastward of the SST front (labeled "2" in Fig. 2). In addition, a local minimum in EKE increase is observed poleward of the SST front (labeled "3" in Fig. 2). These features are all reproduced in the idealized aquaplanet simulation (see corresponding labels in Fig. 2d). In the Southern Hemisphere, the storm tracks on the aquaplanet shift poleward and tend to intensify. In the Northern Hemisphere and downstream of the idealized SST front, EKE increases, while further poleward and equatorward EKE decreases (see corresponding labels in Fig. 2d). Taken together, the pattern of EKE changes in the aquaplanet simulation downstream of and at the idealized SST front is reminiscent of the changes projected by the more complex Earth system models.

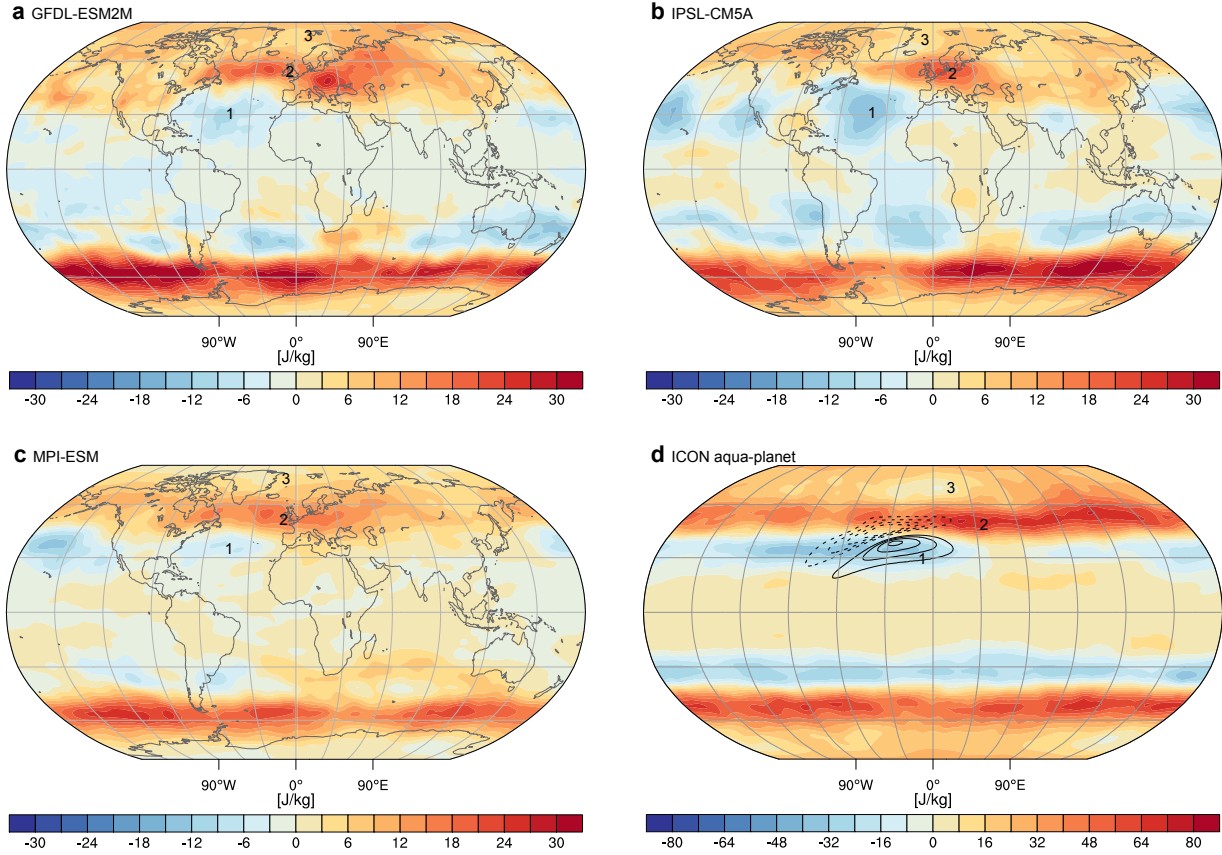

**Figure 2.** Eddy kinetic energy difference (shading; J kg$^{-1}$) at the 250 hPa level between the periods 2091–2100 and 2011–2030 (Dec–Feb) for the RCP.8.5 scenario and for the (a) GFDL-ESM2M (b) IPSL-CM5A LR (c) MPI-ESM LR (d) and an idealized ICON aquaplanet simulation including a SST front in the Northern Hemisphere, which is warmed by 4 K.





In contrast to the Gulf Stream SST front, the EKE change over the Kuroshio is less pronounced (Figs. 2a,b and c), possibly related to climatologically weaker SST gradients over the western North Pacific, but the change pattern is also not absent. The stronger signal over the North Atlantic suggests that likely other mechanisms are at play, including the formation of the North
Atlantic warming hole. The North Atlantic warming hole describes the counter intuitive cooling of the SSTs of the northern North Atlantic in simulations of global warming (Keil et al., 2020). This regional SST cooling has been shown to contribute to locally enhanced near-surface baroclinicity and stronger downstream EKE and eddy momentum convergence (Gervais et al., 2018). However, the aquaplanet simulation shows that the presence of an SST front alone is sufficient to simulate a response of the storm tracks to warming, similar to that found over the North Atlantic in CMIP simulations. In what follows, we focus
our analysis on the mechanisms that lead to the intensification of the storm track downstream of the idealized SST front. This focus does not imply that the North Atlantic warming hole is not an important factor affecting the net response of the North Atlantic storm track to global warming, nor does it mean that the mechanisms causing the response on the aquaplanet are the same as those causing the response in CMIP simulations.

### 3.2 Detailed analysis of the EAPE budget

In the following, we consider the change in EAPE between the +4 K warmed and control simulations, since the EAPE changes may eventually precede those in EKE in both time and space. Unless stated otherwise, fields shown in the following are mass-weighted vertically averaged between 1000–200 hPa.

The poleward shift of the zone with the highest EAPE is clearly seen in both hemispheres from the difference of EAPE between both simulations (shading in Fig. 3a). Further, there is a clear increase in the EAPE magnitude downstream and north-
230 east of the SST front and an eastward expansion of the region with enhanced EAPE. These changes are not found at any other latitude or longitude (red contours in Fig. 3a). The increase is most pronounced at levels above 500 hPa and largest at 300 hPa, while at the 700 hPa and below there is mild to no decrease in EAPE (not shown). The zonal mean of the vertically averaged difference shows that EAPE decreases within a band between 30 and 45°N, but increases poleward of 45°N with a peak at 55°N. Furthermore, there is a locally reduced increase between 60–70°N and another enhanced increase near the poles north
of 75°N in both hemispheres (Fig. 3a).

The differences between both hemispheres are well seen in the meridional mean changes of external baroclinic conversion (Fig. 3b). For the Northern Hemisphere, the difference between the meridional means of both simulations shows an EAPE increase between 0-120°E, which is well above the increase seen in the Southern Hemisphere (black dashed contour in Fig. 3b). Further downstream and upstream, between 120–180°E and 180–0°W, meridionally averaged EAPE in the warmed simulation
is only marginally larger compared to the control simulation (black solid contour in Fig. 3b). However, it is smaller compared to the increase observed in the Southern Hemisphere. This suggests less eddy activity far up- and downstream of the SST front compared to storm tracks that form in the absence of a SST front.

To summarize, the changes in the Southern Hemisphere (black dashed contour in Fig. 3b) indicate that EAPE not only shifts poleward in the warmer climate but increases at nearly all longitudes, which appears to be due to the increase in EAPE



everywhere poleward from 45°N toward the poles (Fig. 3b), while in the Northern Hemisphere the increase is confined but clearly increased downstream of the SST front.

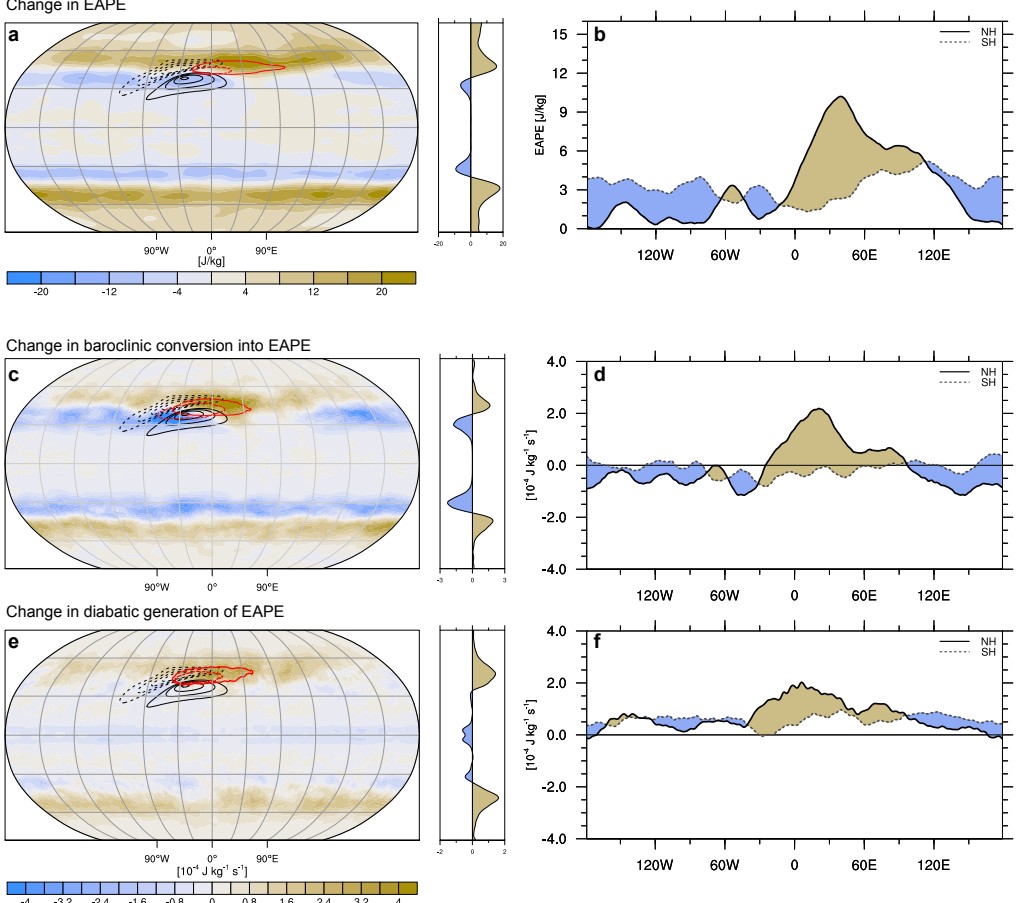

**Figure 3.** (a) EAPE difference between warmed and control simulations (mass-weighted vertically averaged between 1000–200 hPa). Additional red contours indicate selected EAPE value of $90 \, \mathrm{J \, kg^{-1}}$ in the control (red dashed) and warmed (red solid) simulations; (c) Change in external baroclinic conversion and (e) diabatic generation rates of EAPE. Additional red contours in (c) and (e) indicate $8 \times 10^{-4} \, \mathrm{J \, (kg \, s)^{-1}}$ in the control (red dashed) and warmed (red solid) simulations. (b) EAPE difference between the warmed and control simulations but meridionally averaged over the Northern (solid) and Southern (dashed) Hemispheres' midlatitudes (20°–60°N). Color shading highlights differences between the two hemispheres. (d) and (f) as (b) but for external baroclinic and diabatic EAPE generation.

Next, consideration is given to the two leading source terms of EAPE, which are the external baroclinic conversion and diabatic EAPE generation. The maximum EAPE generation by external baroclinic conversion occurs northeast of the SST front (Fig. 3c). As for EAPE, the maximum in external baroclinic conversion is shifted north-eastward relative to the SST front

(Fig. 3c), and the absolute values clearly increase in the warmed simulation (red contours in Fig. 3c). The difference between





the meridionally averaged midlatitude (20–60°N) external barolinic conversion rates shows in the Northern Hemisphere a clear increase between 30°W –90°E (Fig. 3d), which is most pronounced downstream of the SST front between 30°W–60°E. It agrees well with the EAPE change (Fig. 3b). In fact, the peak is slightly upstream of the increase in EAPE indicating the importance of EAPE advection. At all other longitudes, the meridionally averaged external baroclinic conversion in the NH is

reduced (solid contour in Fig. 3d) relative to the control as well as relative to the change in the SH (blue shading in Fig. 3d). In fact, in the Southern Hemisphere, the difference in the meridionally averaged conversion fluctuates around zero (dashed contour in (Fig. 3d), indicating that the poleward shift of the external baroclinic conversion in the warmer climate is not accompanied by an increase in the external baroclinic conversion (this also holds true if the conversion rate is meridionally averaged over the entire hemisphere instead of midlatitudes only). It also indicates no increase in EKE from enhanced baroclinic conversion.

However, in the zonal mean the change in external baroclinic conversion shows a dipole structure akin to the shift of the storm track. The positive and negative peaks largely cancel each other. This implies that the northern hemispheric local increase of external baroclinic conversion downstream of the SST front is largely balanced by a decrease elsewhere at the same longitude. Consequently, the *local* increase in EAPE downstream of the SST front is partly explained by increased external baroclinic conversion but the *global* increase in EAPE seen in Figs. 3a,b is not.

Before turning attention to the diabatic EAPE generation, we take a closer look at the baroclinic conversion changes near the SST front. External baroclinic conversion into EAPE is the product between the eddy total energy, the mean baroclinicity and the conversion efficiency. In response to the imposed uniform surface warming, the mean baroclinicity shifts poleward (black contours in Fig. 4). The shift in the SH is mostly symmetric, while in the NH the poleward increase in mean baroclincity is zonally limited to an elongated band extending from upstream of the SST front to approximately 120°E, while there is no

increase further downstream. At the same time, there is a reduction up- and downstream of the warm side of the front but only a reduced weakening in the direct vicinity downstream of the SST front, where there is a local increase in conversion efficiency. Eddy total energy, the sum of EKE and EAPE, increases as expected downstream and to the north-east of the SST front, indicating the stronger storm track (red contours in Fig. 4). Much of the local change in external baroclinic conversion is dictated by local changes in the conversion efficiency (shading in Fig. 4), which increases poleward of the SST front but in par-

ticular to the northeast and downstream of it. The external baroclinic conversion increases at all analyzed levels above 500 hPa while there is a weakening below 700 hPa (not shown). As the external baroclinic conversion does not change globally, the SST front in the first place acts to localize the external baroclinic conversion downstream of the front. Apparently to the north of the SST front and downstream the re-organization by the SST front in the warmer climate is such that external baroclinic conversion efficiency increases, which indicates a better alignment of the eddy heat flux with the mean baroclinicity compared

to the control. As a consequence, the eddies become more efficient in tapping into the mean potential energy reservoir.

The second main EAPE source term is the diabatic generation. The change of the net diabatic EAPE generation, which in absolute terms is strongest near the SST front (red contour in Fig. 3e), also indicates a poleward shift in both hemispheres (Fig. 3e). The difference between the meridional means indicates an increase in the NH near the front (solid contour in Fig. 3f) that exceeds the increase in the SH at the same longitude (brown shading in Fig. 3f). However, at other longitudes, the

increase in the meridional mean is generally weaker in the NH compared to the SH (east of 100°E and west of 50°W; Fig. 3f).



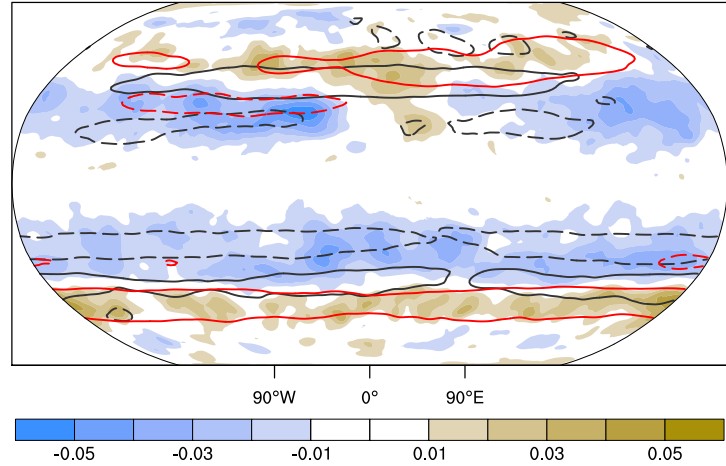

**Figure 4.** Change in external baroclinic conversion efficiency (shading), eddy total energy (red; $\pm 25 \times$ J kg) and background baroclinicity (black contours; dashed indicate negative and solid positive values; units of $\pm 1.5 \times 10^{-6}$ J kg$^{-1}$). All field are mass-weighted vertically averaged between 1000–200 hPa.

This is further underlined by comparing the meridionally averaged changes in the two hemispheres, which are positive almost everywhere (see dashed and solid contours in Fig. 3f). Consequently, the zonal means are positive poleward of 45°N and 45°S and mostly symmetric between the two hemispheres. Hence, we conclude that *globally* the increase in EAPE (Fig. 3b) is primarily explained by enhanced diabatic generation of EAPE.

In the following, the diabatic EAPE generation is partitioned into the contributions of different diabatic processes (Fig. 5). Even though the diabatic EAPE generation depends on the underlying model, the sub-grid scale parameterizations and their specific design, a decomposition of the diabatic EAPE tendency is still of scientific merit as it points towards the dominant physical processes accounting for the changes. The strongest increase results from the resolved-scale condensation and the cloud microphysics scheme, which simulates in-cloud phase changes of water, such as ice formation and sublimation (Fig. 295    5a). In the NH there is a clear increase of diabatic generation of EAPE by microphysical processes north of the front. This increase extends from upstream near 90°W of the front and downstream to 120°E. However, the zonal mean indicates that much of the increase is globally compensated by a decrease further equatorward. The change in resolved-scale condensation and evaporation, i.e., due to saturation adjustment, is of similar importance near the SST front in terms of magnitude. However, the absence of a clear dipole structure in the zonal mean indicates that the global increase in resolved-scale condensation and 300    evaporation is stronger compared to that resulting from the cloud microphysics parameterization (Fig. 5b).

The diabatic EAPE tendencies from the other processes are considerably smaller. EAPE longwave radiation tendencies change sign at different levels (Fig. C1). At the 700 and 500 hPa level EAPE tendency from longwave radiation is positive, which are levels where the cloud base is warmed from thermal radiation received from the earth surface (Fig. C1). Thus, in the vertically averaged framework, the positive and negative anomalies in Fig. 5c result not only from a poleward shift but also





from an enhancement of the existing positive and negative patterns originating from different vertical levels. Nevertheless, the magnitude is considerably smaller compared to that of cloud physics and condensation and the zonal mean suggests only little global change. In contrast to the longwave radiation, the EAPE tendencies from shortwave radiation partly oppose longwave radiation at upper levels. In general the shortwave EAPE tendencies are close to zero in the lower levels of the tropopshere (e.g., 700 hPa), while they are positive in the upper troposphere (e.g., 300 hPa), thereby partially counteracting the negative longwave

tendencies (Fig. C1). EAPE generation by turbulence is negative north of the front and positive south of it, weakening with height (Fig. 5e). Turbulence contrasts with EAPE tendencies from parameterized convection, which includes turbulent mixing in convective plumes and latent heating. Convective EAPE generation increases at the SST front (Fig. 5f) and poleward of the maximum in the control run (Fig. C1b), with no concurrent and well-marked reduction equatorward (Fig. 5f).

In summary, the local increase in external baroclinic conversion rates dominates the strong local increase in EAPE down-

315 stream of the SST front with only a smaller contribution from diabatic EAPE production, which is slightly increased near the SST front. The global EAPE increase, however, is the result of increased diabatic EAPE production in the first place due to resolved-scale condensation.

### 3.3 Detailed analysis of the EKE budget

Following the discussion of the EAPE budget, this section discusses the EKE change shown in Fig. 6a, the budget and related

changes. The EKE change in the NH exhibits a strong asymetry between the region upstream and downstream of the SST front (Fig. 6b). In the SH, EKE is solely shifted poleward with no notable increase. The two main EKE source terms are the internal baroclinic conversion, which converts EAPE into EKE (third term on the r.h.s. in Eq. 6), and the barotropic conversion (fourth term on the r.h.s. in Eq. 6), which generates EKE from the low-frequency background flow. The ageostrophic geopotential fluxes and advection are not discussed as both processes are no global EKE sources or sinks but redistribute EKE locally.

The vertically averaged time-mean barotropic conversion is mostly negative in our simulations (not shown) and acts as an EKE sink as it is generally the case within storm tracks (Chang et al., 2002). The change in EKE, shown in Fig.6a, displays the aforementioned tripole with an EKE decrease equatorward and uptream of the SST front's center, an increase poleward and downstream, and a slight reduction at the longitude of the SST front's center but at high latitudes (at 30°W ,70°N in Fig.6a. The meridionally averaged (20–60°) change in EKE is for the SH close to zero (dashed contour in Fig.6b), which once again

indicates that the SH experiences a poleward shift with not marked increase in EKE in the midlatitudes. EKE increases if averaged over the entire extratropics (20–90°), but this increase is dominated by EKE at polar latitudes with no to little change in midlatitudes. In the NH, EKE reduces upstream and equatorward of the front between 0° and 180° and increase downstream and poleward of the front (red solid contour in Fig.6a). The storm track hence shifts poleward and becomes weaker upstream of the front relative to the control run and stronger downstream of the front.

As is the case for external baroclinic EAPE generation, there is a distinct increase in in internal baroclinic generation of EKE (Fig.6c). The increase in internal baroclinic conversion into EKE starts, similar as for external baroclinic conversion into EAPE, upstream of the front, reaches a first maximum near 25–30°W and continues to be increased downstream with a second maximum near 60°W, which is best seen in the meridional average (Fig.6d). The increase exceeds the change seen in the SH.



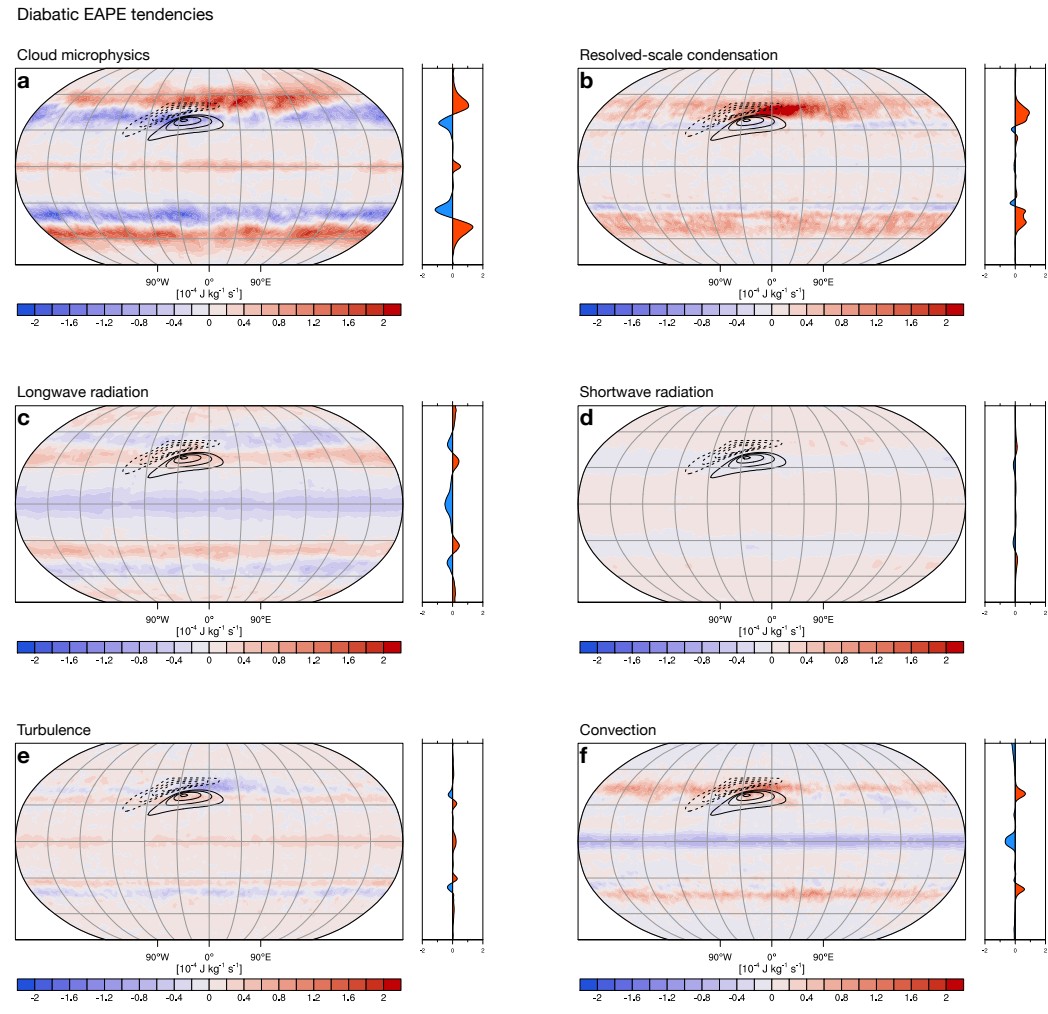

**Figure 5.** Mass-weighted vertically averaged change of different diabatic EAPE tendencies between the warmed and control simulation due to (a) cloud microphysics, (b) saturation adjustment, (c) longwave and (d) shortwave radiation, (e) turbulence and (f) parameterized convection (color shading). Vertical averages are taken between 1000–200 hPa.

Further downstream and around the dateline, baroclinic conversion into EKE is reduced compared to the levels seen in the
SH. Overall, the change in the meridionally averaged internal baroclinic conversion is small in the SH indicating a poleward shift with only a minor net increase (the dashed line in Fig.6d is mostly above the zero contour). The change in the barotropic tendency, which is climatologically negative, is positive in the area of and slightly upstream from the SST front (20°W–5°E), indicating a lower barotropic EKE sink in this area. Downstream, between 60°–150°W, the negative barotropic EKE tendency is increased, which indicates enhanced transformation of EKE to zonal mean kinetic energy. For the SH, the meridional average
of the change is close to zero, indicating a poleward shift with no marked change in amplitude. To summarize, as for EAPE it



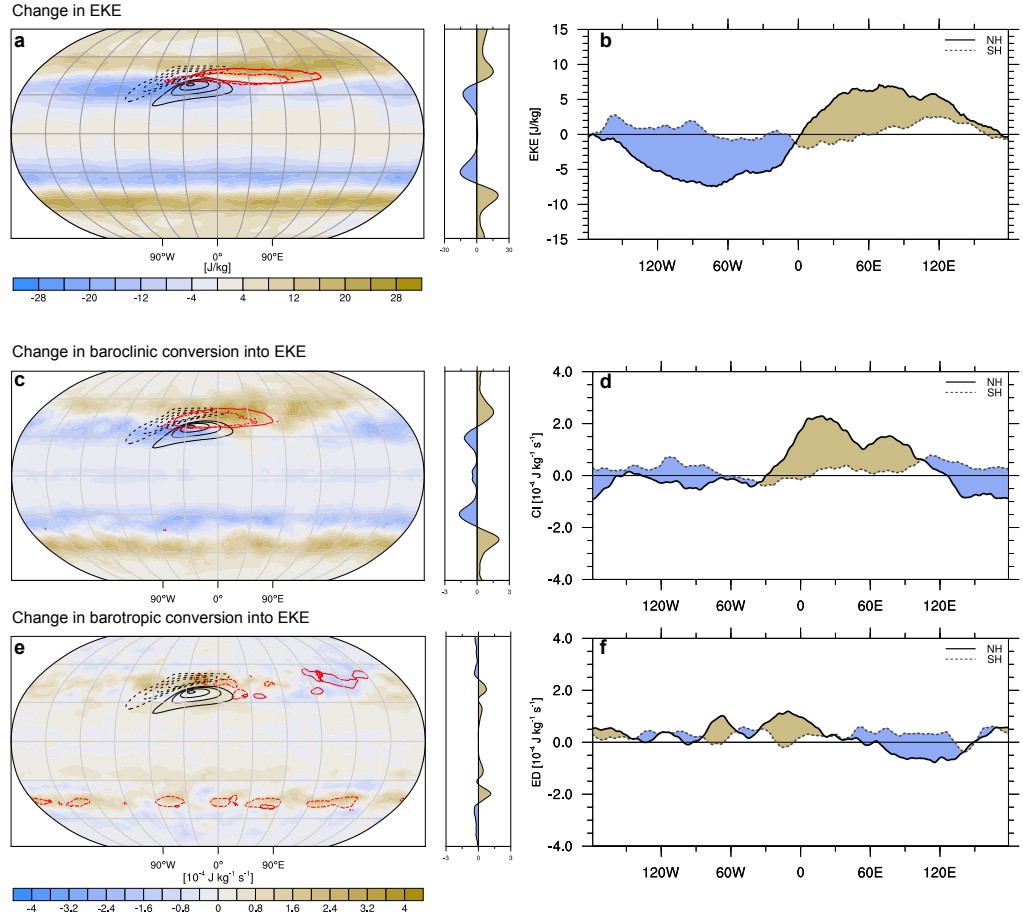

**Figure 6.** (a) EKE difference between warmed and control simulations (mass-weighted vertically averaged between 1000–200 hPa). Additional red contours indicate selected EKE value of $135 \, \mathrm{J \, kg^{-1}}$ in the control (red dashed) and warmed (red solid) simulations; Black contours indicate the SST front. (c) Change in internal baroclinic and (e) barotropic generation rates of EKE. Additional red contours in (c) indicate $12 \times 10^{-4} \, \mathrm{J \, kg^{-1}} \, s$ in the control (red dashed) and warmed (red solid) simulations and in (e) indicate $-3 \times 10^{-4} \, \mathrm{J \, (kg \, s)^{-1}}$ (Note that the time mean barotropic conversion is mostly negative). (b) EKE difference between the warmed and control simulations but meridionally averaged over the Northern (solid) and Southern (dashed) Hemispheres' midlatitudes ($20°$–$60°$). Color shading highlights differences between the two hemispheres. (d) and (f) as (b) but for baroclinic and barotropic EKE generation.

is primarily the internal baroclinic conversion from EAPE into EKE that drives the observed EKE change downstream of the SST front.





## 4 Change in characteristics of extratropical cyclones from feature-based tracking

In the following section, consideration is given to the changes in several life cycle characteristics of extratropical cyclones
obtained from a featured-based surface cyclone tracking. The total number of analyzed cyclone tracks from the 20 winter seasons is approximately 15'900 in the control simulation and 13'500 in the warmed simulation, corresponding to a reduction by approximately 15 %, which is in agreement with earlier studies (Sinclair et al., 2020).

### 4.1 Change in cyclone frequency

The cyclone frequency indicates the relative proportion of all time steps at which a grid point is affected by a surface cyclone.
The frequency is climatologically high in regions where cyclones are in their mature phase (and cover a wide area) and in regions where the propagation speed is low (they affect the same region during several time steps). Such regions are typically located downstream and poleward of high EKE regions. High EKE indicates regions of high cyclone intensity and propagation speed (Schemm and Schneider, 2018).

Figure 7 shows the change in surface cyclone frequencies alongside the 20 % absolute cyclone frequency contour. Consistent
with the changes in EAPE and EKE, high cyclone frequencies shift poleward in both Hemispheres. In the zonal mean, the increase on the poleward side is spread over a wide band of latitudes, while the decrease on the equatorward side affects a smaller band of latitudes but reaches higher amplitude. Downstream of the SST front, the increase is confined to a region to the northeast of the front near 90–120°E. Further downstream and far upstream, the frequency of cyclones decreases in the warmer simulation without being accompanied by an increase poleward of similar strength. These are regions that are also
marked by a decrease of EAPE and external as well as internal baroclinic conversion in the warmed simulation, indicating an earlier termination of the intensified storm track. As shown below, this is in agreement with a reduced cyclone life time. Earlier termination of the storm track, accompanied by shortened cyclone lifetimes are features observed under present-day climate conditions during mid-winter over the North Pacific, in particular for cyclones originating downstream of the Kuroshio SST front (Schemm et al., 2021).

### 370 4.2 Change in lifetime

The frequency distribution of the lifetimes of surface cyclones that have minimum lifetimes greater than 1 day is shown in Fig. 8a along with the median, 95[th], 99[th], and 99.9[th] percentiles on the lower axes (red dots correspond to percentiles in the warmer simulation, black dots to the control simulation). Lifetimes decrease in the warmer simulation for lifetimes greater than 3 days. While the median lifetime remains unchanged (red and black dots at 2.2 days), the lifetime tends to decrease for
all percentiles greater than the 90[th] percentile (Fig. 8b).

For cyclones originating near the SST front, the distribution of lifetimes is skewed towards longer lifetimes compared to all cyclones (Fig. 4c). While the lifetime of the median cyclone remains unchanged between the control simulation and the simulation with additional warming, the lifetime tends to increase for the high percentiles. For the 99.9[th] percentile, for





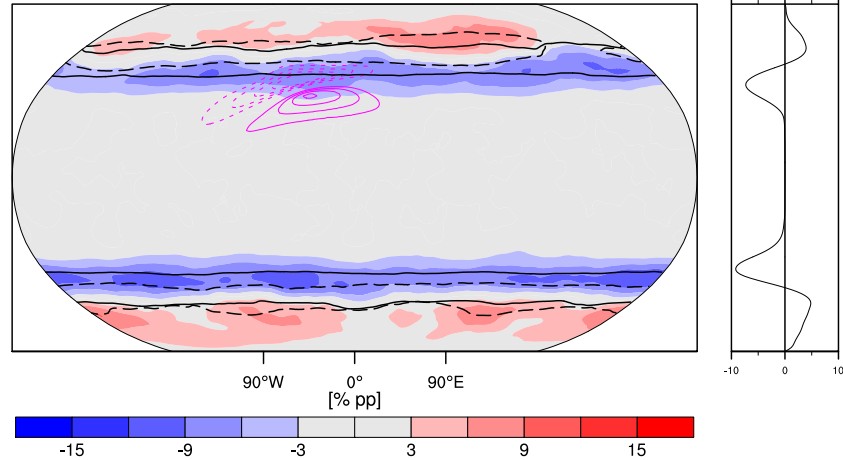

**Figure 7.** Change in surface cyclone frequencies (shading; percentage points) between warmed and control simulations. Additional black contours indicate absolute cyclone frequency of 20 % (solid for the warmed simulation; dashed for the control. As in the previous figures, the SST front is indicated by purple contours.)

example, it increases by more than 24 hours. However, the upward trend alternates between bins and thus appears to be less
robust compared to the general trend toward shorter lifetimes in the warmer simulation.

### 4.3   Change in deepening rates

The frequency distribution of cyclone deepening rates, calculated based on all 6-h minimum SLP changes along all cyclone tracks, has a median value of about $2\,\text{hPa}\,(6\,\text{h})^{-1}$ and is skewed towards larger deepening rates, with the 99th percentile in the control simulation at $10\,\text{hPa}\,(6\,\text{h})^{-1}$ (Fig. 9a). For cyclones originating near the SST front, the distribution of deepening rates
is skewed toward even higher values; for example, the 99th percentile is $11\,\text{hPa}\,(6\,\text{h})^{-1}$ or higher by about half the median value (Fig. 9c). The frequency of very high deepening rates increases in the warmer simulation, while the frequency of deepening rates below the median of the control simulation decreases (Fig. 9b, d). Consequently, the high percentiles shift toward higher deepening rates, e.g., the 99th percentile decreases by about $1\,\text{hPa}\,6\,\text{h}^{-1}$ for all cyclones and for those with genesis near the SST front, while the median changes only slightly.

### 390   4.4   Change in maximum intensity

Maximum intensity is defined as the lowest SLP value found along a track and based on each value from all tracks the frequency distribution in Fig. 10a is computed and for all tracks with genesis near the SST front in Fig. 10b. The distributions are normally distributed with median values in the control simulation approximately at $990\,\text{hPa}$ and the 99th found at $970$ hPa (Fig. 10a,b). In the warmer simulation, the maximum intensity shifts towards lower pressure values, in particular for the very deep cyclones,
the 99.9th increases from $945\,\text{hPa}$ to approximately $935\,\text{hPa}$ (Fig. 10a). Overall, the median cyclone intensity in the control

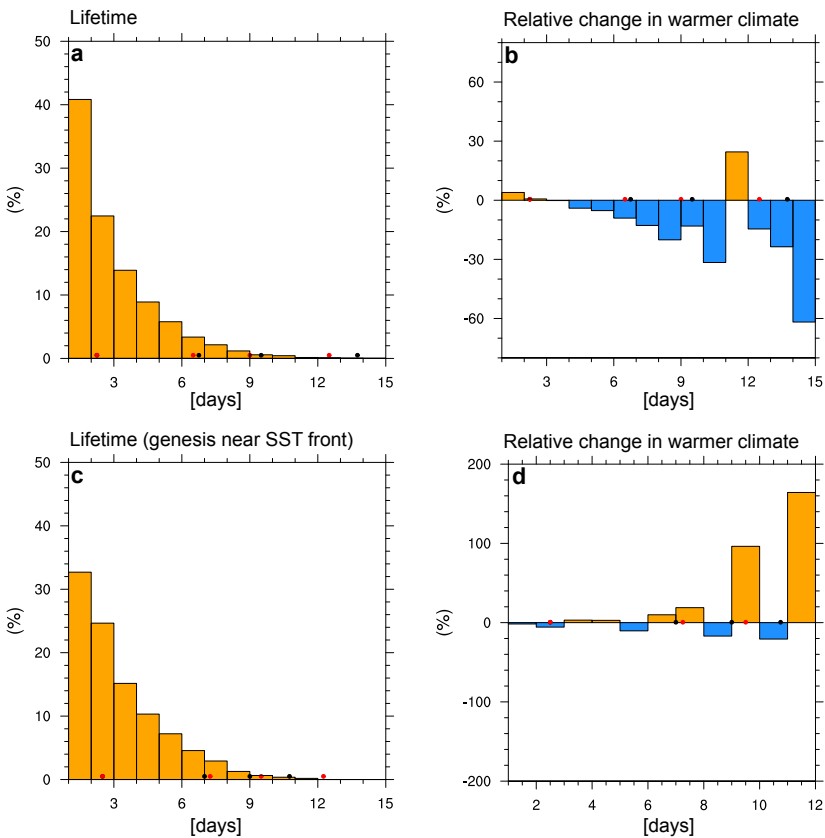

**Figure 8.** Frequency distribution of lifetimes of extratropical cyclones for (a) all cyclones (NH and SH) and (c) cyclones with genesis in the vicinity of the SST front in the control simulation. On the lower axes, dots indicate the median, the 95th, the 99th and the 99.9th percentiles for the control (black) and +4 K (red) simulation. (b,d) Relative change of the distribution in the +4 K simulation. Note that the large relative change for the high percentiles results from the low number of cases in these bins.

simulation becomes less frequent, while the frequency of very deep cyclones increases and this increase is not limited to those cyclones with genesis near the SST front.

### 4.5 Change in poleward propagation

In the following, we consider the poleward displacement of surface cyclones, which is defined as the meridional difference
between genesis and lysis or genesis and the latitude of maximum intensity, respectively. The latter, in turn, is defined as the latitude at which the SLP minimum is observed along a track. The subsequent analysis is motivated by the reported enhanced poleward propagation of cyclones in a warmer climate (Tamarin and Kaspi, 2017; Tamarin-Brodsky and Kaspi, 2017). Poleward propagation of surface cyclones, which largely depends on a nonlinear mechanism, increases with the amplitude of lower- and upper-level disturbances (Gilet et al., 2009; Oruba et al., 2013). Consistently, the regions of high cyclone intensity are located



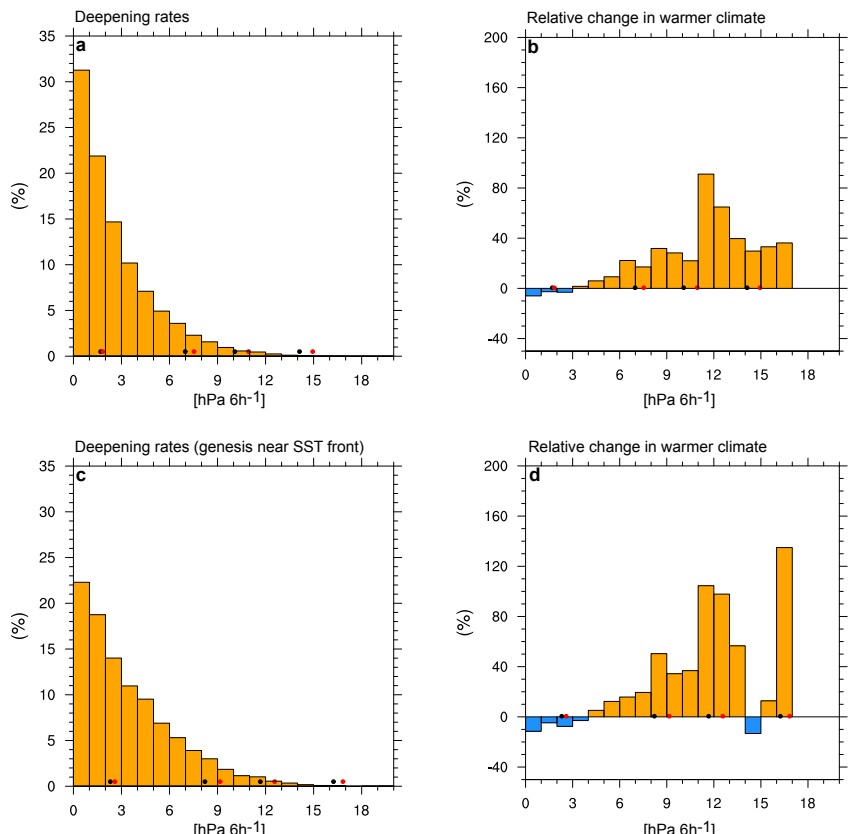

**Figure 9.** Frequency distribution of deepening rates, defined as 6-hourly minimum SLP changes, for (a) all cyclones and (c) cyclones with genesis in the vicinity of the SST front in the control simulation. On the lower axes, dots indicate the median, the 95[th], the 99[th] and the 99.9[th] percentiles for the control (black) and +4 K (red) simulation. (b, d) Relative change of the distribution in the +4 K simulation. Note that the large relative change for the high percentiles results from the low number of cases in these bins.

where the strongest poleward motion occurs over the main ocean storm tracks (Besson et al., 2021). In a warmer climate, the increased moisture intensifies upper-level potential vorticity anomalies that more easily advect surface cyclones poleward (Coronel et al., 2015; Tamarin and Kaspi, 2016). Additionally diabatic low-level PV generation eastward of the surface cyclone promotes intensification of the surface cyclone and the interaction between the diabatically-induced circulation and the cyclone also advects the cyclone poleward.

These earlier results suggest that cyclones with maximum deepening rates downstream of the SST front, which have been shown to be higher than elsewhere, exhibit advanced poleward displacement compared to cyclones elsewhere. The meridional displacement between cyclogenesis to cyclolysis, computed as the mean over all displacements is 4.2° latitude in the control run and 4.3° latitude in the warmer run but under an on average reduced lifetime. A clearer picture emerges when considering only cyclones with maximum deepening rates downstream and in the vicinity of the SST front. The averaged displacement



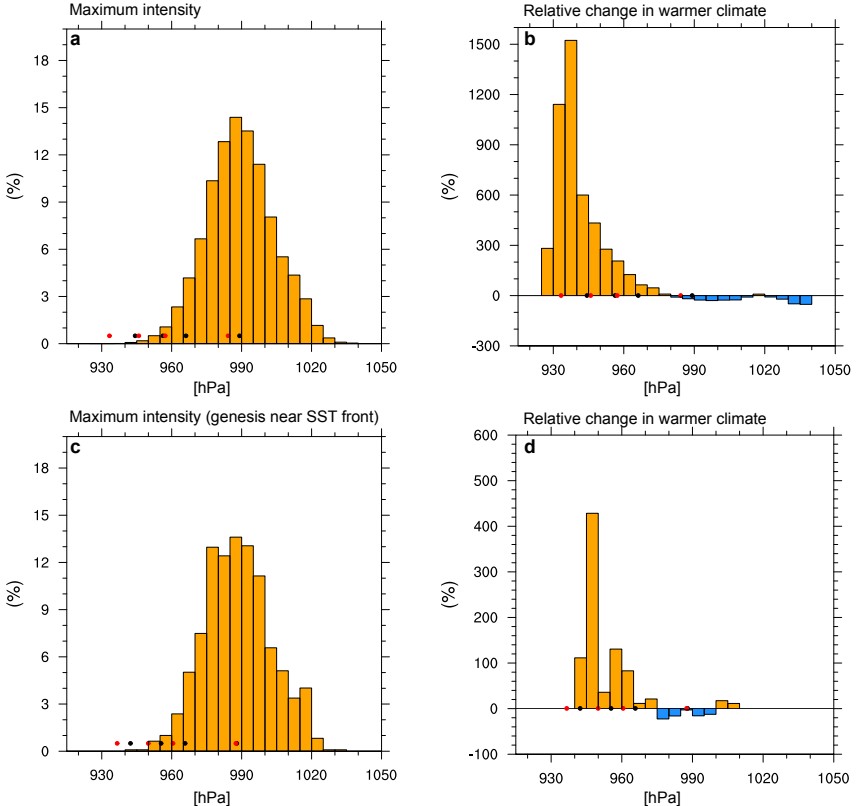

**Figure 10.** Frequency distribution of maximum intensity, defined as minimum SLP during the entire life cycle, for (a) all cyclones and (b) cyclones with genesis in the vicinity of the SST front in the control simulation. On the lower axes, dots indicate the median, the 95th, the 99th and the 99.9th percentiles for the control (black) and +4 K (red) simulation. (b,d) Relative change of the distribution in the +4 K simulation.

increases from 7.4° latitude in the control simulation to 8.4° latitude in the warmer simulation, in agreement with expectations. Here, it is in particular the poleward shift in lysis that dominates the change (56.5°N in the control and 59°N in the warmer simulations) because cyclogenesis is anchored near the SST front in both simulations. When the displacement is considered between cyclogenesis and maximum intensity, for all cyclones we find this to be on average 1.3° latitude in the control run (downstream of the SST front 1.5° latitude) and 1.0° latitude in the warmer run (downstream of the SST front 1.1° latitude).

This appears at first to disagree with the results of Tamarin and Kaspi (2017) and Tamarin-Brodsky and Kaspi (2017). However, limiting the selection to the 200 deepest cyclones downstream of the SST front, which become even deeper in the warmer simulation and live longer, the poleward shift increases from 1.7° latitude in the control simulation to 2.3° latitude in the warmer simulation. Thus, while the majority of cyclones exhibits only marginal change in the poleward displacement until maximum intensity, the opposite is true for the strongest cyclones, which is also the selection criterion used by Tamarin and

Kaspi (2017) and Tamarin-Brodsky and Kaspi (2017) and thus in agreement with these studies.



## 5 Conclusions

This study examines the response of the storm track downstream of an idealized SST front to uniform global warming by +4 K on an aquaplanet. The pattern of change reveals a tripole structure that has received little attention to date but is also seen in CMIP5 simulations under the RCP 8.5 scenario when comparing present-day to end-of-century conditions over the North Atlantic and partly over the North Pacific. The hemisphere on the aquaplanet that includes the idealized SST front successfully reproduces the tripole pattern, which consists of an EKE decrease upstream and equatorward of the front, an EKE increase downstream and northeast of the front, and a regional local minimum in the EKE increase at polar latitudes. The hemisphere without the SST front exhibits a rather zonally symmetric poleward shift. A detailed analysis of the EAPE and EKE budgets shows that the change pattern is explained locally by increased or decreased external and internal baroclinic conversions, respectively. The SST front organizes the flow such that the baroclinic growth becomes more efficient downstream of the front, that is, the eddy heat flux better aligns with the baroclinicity vector. However, the hemisphere-wide change in baroclinic conversions is close to zero because it is reduced far downstream and upstream of the SST front compared to the control simulation. Notably, the SST front re-organizes and localizes baroclinic conversion without changing it globally and thus can be thought of the chief organizer behind the spatial baroclinic conversion pattern. In the absence of a front, i.e., in the southern hemisphere in our simulations, the midlatitude meridional average of the changes of baroclinic conversions is thus also close to zero, indicating a poleward shift with little change in net conversion rates. Instead, the global increase in EAPE is explained by enhanced diabatic EAPE generation, primarily due to parameterized cloud physics and, in particular, resolved condensation. Diabatic EAPE tendencies due to longwave radiation are negative in the upper levels and positive in the lower levels and are partially offset by shortwave radiation. Turbulent generation of EAPE is partially offset by parameterized convection, especially near the SST front, with both changing sign over the front but in opposite directions.

A feature-based surface cyclone detection scheme is used to detect changes in life cycle characteristics. Broadly consistent with previous idealized studies (Sinclair et al., 2020), we find a decrease in mean cyclone intensities and mean deepening rates, as well as an increase in strong cyclone intensities and strong deepening rates. This corresponds to a flatter probability density function of the two quantities in a warmer climate with a longer tail towards extremes. A reduction in mean lifetime is also noted, with the exception of the cyclones that form downstream of the SST front, which live longer in a warmer climate. The poleward displacement, defined as the latitudinal shift between genesis and lysis, remains fairly unchanged when averaged over all cyclones, but increases for cyclones with genesis downstream of the SST front. Defined as the latitudinal shift between genesis and location of maximum intensity, instead of lysis, the poleward shift even decreases when averaged over all cyclones but increases for the subset of 200 strongest cyclones, which is consistent with Tamarin and Kaspi (2017). Again, the probability density function becomes flatter with longer tail. The most notable difference between all cyclones and those with genesis at the SST front is the opposite response of their lifetimes to warming. Finally, it is noted that the number of cyclones decreases by about 15 %.





## 5.1 Limitations and future direction

The main limitation of this study is its idealized nature. The ocean acts as infinite source and sink of heat, which is not balanced
by ocean heat uptake elsewhere. A slab ocean would be a useful extension to our setup. Consideration is given in this study to
the main EAPE and EKE source and sink terms, but a study on the redistribution of EAPE and EKE by changes in advection
and ageostrophic flux around the SST front (Orlanski and Katzfey, 1991) would be interesting to further elucidate the changes
in the internal structures of the storm track such as the downstream development.

Future studies may also probe the sensitivity of the formation of the EKE tripole pattern to the shape, strength and orientation
of the imposed idealized SST front and to the used model resolution. A detailed analysis of the sensitivity of the baroclinic
conversions, its efficiency and the vertical tilt of eddies at the SST front, as well as the sensitivity to the specifics of the SST
front (shape, orientation and strength) is left for future research. Further studies may also consider changes in special cyclone
types, such as frontal-wave cyclones (Schemm and Sprenger, 2015) or diabatic Rossby waves (Boettcher and Wernli, 2013).



## Appendix A: Initial SST profile

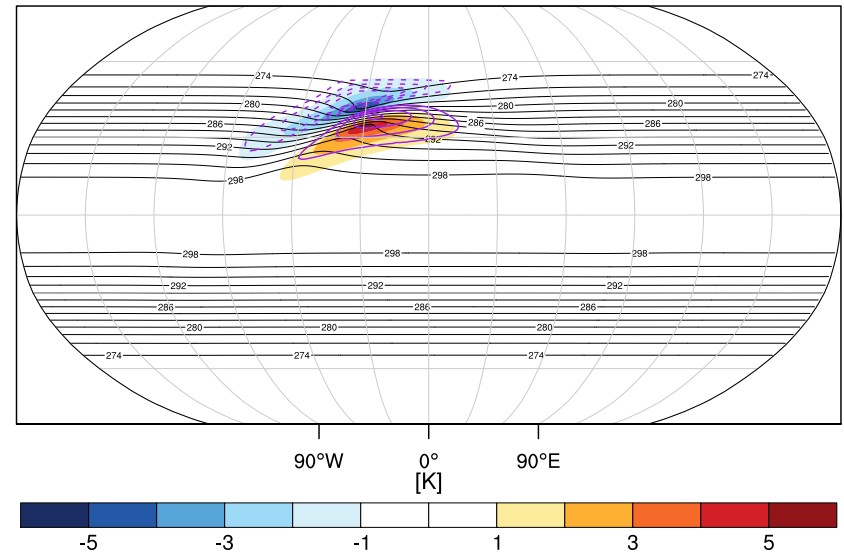

**Figure A1.** Zonally symmetric initial SST profile known as "Qobs" (black contours), additional SST anomaly used to construct the SST front (colored shading), and resulting 2-m temperature anomaly calculated as a deviation from the climatological zonal mean 2-m temperature (purple contours with solid indicating positive and dashed negative values; $\pm 1°, 1.5°, 2°$ and $2.5°$). The latter is shown in all figures in the manuscript.




## Appendix B: Zonal mean change

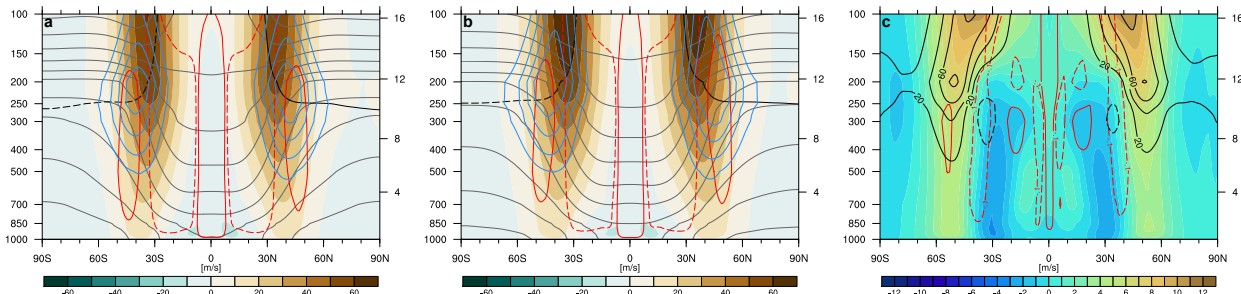

**Figure B1.** Zonal mean fields in the (a) control and (b) control+4K simulations and (c) their difference for the zonal wind speed (shading; m s$^{-1}$), vertical wind speed (red contours, negative dashed; cm s$^{-1}$) and eddy kinetic energy (blue contours in (a,b) and black contours in (d); J kg$^{-1}$ s$^{-1}$). Additionally shown in (a) and (b) is the dynamical tropopause (black contour; 2 PVU) and potential temperature (gray contours).

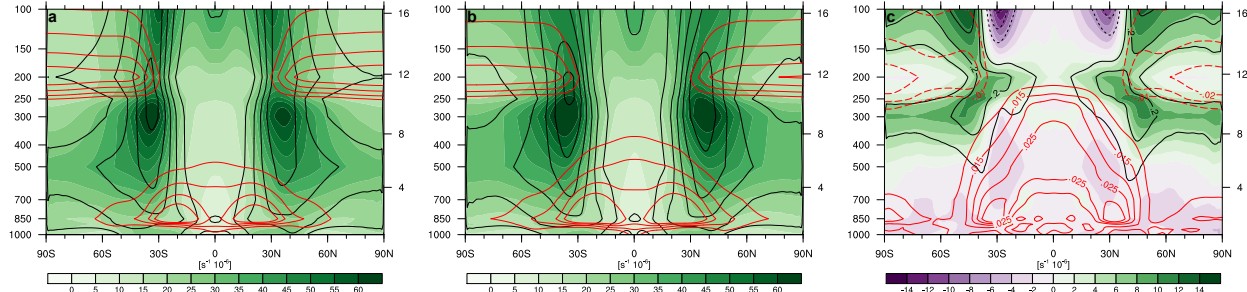

**Figure B2.** Zonal mean baroclinicity (green shading; s$^{-1}$10$^{-6}$), meridional potential temperature gradient (black contours; K 100km$^{-1}$) and the dry static stability normalized by the scale height (red contours; K$^2$ m$^{-2}$ s$^2$) in the (a) control and (b) control+4K simulations and (c) their difference.





## Appendix C: Individual diabatic EAPE tendencies in the control simulation

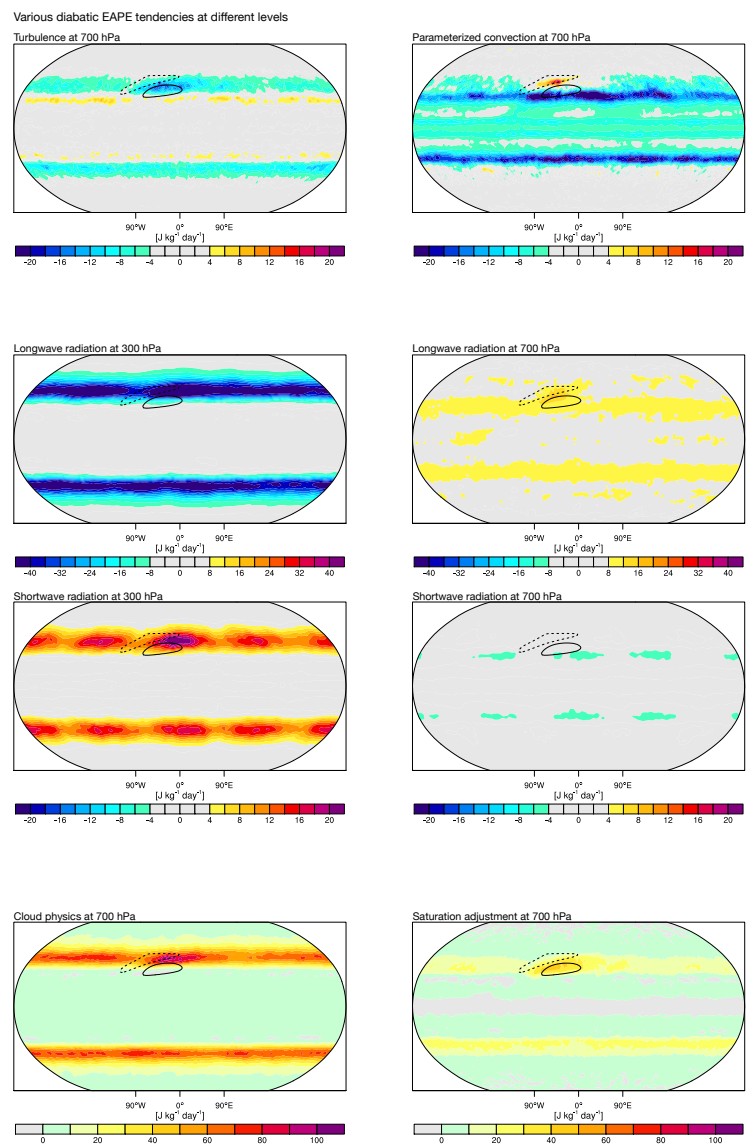

**Figure C1.** Diabatic EAPE tendencies for different processes on different selected levels for the control simulation. Note the different scaling for each panel. EAPE tendencies due to (a) turbulence and (b) parameterized convection at 700 hPa; Longwave radiation at (c) 300 and (d) 700 hPa; Shortwave radiation (e) at 300 and (f) 700 hPa (Note reduced scaling by a factor of two compared to longwave radiation); (g) Cloud physics and (f) saturation adjustment at 700 hPa. Units are J kg day$^{-1}$.



*Author contributions.* LP and SeS implemented and performed the ICON aquaplanet simulations. GR and SeS carried out the energy diagnostics. SeS carried out the Lagrangian diagnostics. All authors contributed to the interpretation of the results and the writing of the manuscript. SeS conceived the idea for the study.

*Competing interests.* The authors declare no competing interests.

*Acknowledgements.* The Center for Climate Systems Modeling (C2SM) at ETH Zurich is acknowledged for providing technical and scientific support. All simulations were carried out at ETH's Scientific and High Performance cluster EULER.



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
