# Peer review of "Storm track response to uniform global warming downstream of an idealized sea surface temperature front"

_Weather and Climate Dynamics, 2022_

## Author Comment (AC1)

Reply to reviewer's comments #1

Summary

Using an idealized aquaplanet model, the authors investigate the changes of storm track in response to uniform global warming downstream of a midlatitude SST front. They found a local enhancement of eddy kinetic energy (EKE) downstream of SST front, and attribute the EKE enhancement to the internal baroclinic conversion from eddy available potential energy (EAPE). By analyzing the EAPE budget, they further show that the EAPE enhancement downstream of SST front is due to change of baroclinic conversion efficiency. They also provide the response of several life cycle characteristic of extratropical cyclones using tracking algorithm.

Recommendation

Several prior studies have examined the overall poleward shift of storm track without the presence of diabatic forcing, but less attention has been paid to the local increase of storm track downstream of SST front. I believe this study brings an important contribution to storm track response by considering the local SST-front effect. The results are novel and robust. I thus recommend the authors to perform a minor revision by considering the comments listed below.

**Reply:** We would like to thank the reviewer for the positive evaluation of our study and the helpful comments. In the revised manuscript we have given more emphasis to the local minimum of EKE increase in region 3, which turns out to be largely related to changes in stationary waves, as well as to some extent reduce baroclinic conversion.

Minor comments:

(1) In the abstract, the authors suggest a tripolar pattern of storm track under global warming, with a poleward shift of storm track, enhance EKE downstream of the SST front and a regionally reduced EKE increase at polar latitudes. The local minimum of EKE in polar region in Figure 2 (label 3) is much less obvious than the other two characteristics (labels 1 and 2). Moreover, less attention has been paid to explain this response in the main text. I thus suggest the author to either revise the abstract or add more discussions on the storm track response in the polar region.

**Reply:** The local minimum of EKE in the polar region you are referring to indeed has a weaker amplitude than the anomalies 1 and 2. Nevertheless, it is a common feature of the storm track response that appears in global warming scenarios in various climate models as well as in aquaplanet simulations with a localized SST front. Furthermore, the minimum in region 3 is also evident in 300 hPa wind speeds (Fig. R1a). Hence, we still think it is worth being mentioned in the abstract. However, we do agree that the changes leading to the reductions of EAPE and EKE in region 3 have insufficiently been discussed in the original manuscript.

The reasons for the minimum in region 3 are twofold: Reduced baroclinic conversion and advection anomalies. Considering external and internal baroclinic conversions, we find local minima the region in question (Figs. 3c and 6c). These minima are, however, less pronounced than the minima in EAPE and EKE themselves, suggesting that other processes might be relevant too. A key difference between the warmed and control simulations is the amplification

of a stationary wave pattern downstream of the front. This pattern comprises a strengthened trough downstream and poleward of the SST front, extending about 90° eastward, as well as an amplified ridge between 90°E and 180°E (Fig. R1b). This pattern results in enhanced poleward advection of EKE and EAPE between the stationary trough and ridge (at about 90°E) and reduced advection just north of the SST front. The latter contributes to the existence of the minimum in region 3.

In the revised manuscript, we include Fig. R1 as Fig. D1 and include a new subsection 3.4 discussing the aforementioned changes of stationary waves and their implications for the advection of EAPE and EKE. Furthermore, we also explicitly point out the local minima in external and internal baroclinic conversion in region 3.

[Figure]

*Figure R.1: (a) Wind speed difference at 300 hPa between the Qobs control run with SST front in the Northern Hemisphere and uniform warming by 4 K at the surface, as well as the 40 m s⁻¹ wind speed contour in the control (black) and the warmed simulation (gray). (b) EKE (color) and geopotential difference at 500 hPa (defined as deviation from the the zonal mean; green solid indicate positive and dashed negative values; +/- 200, 400, and 600 m 2 s −2 ). Additionally shown in both panels is the 2-m temperature after removing the zonal mean (black solid contours indicate positive and dashed negative values; starting at +/- 2.5 K in steps of 0.5 K).*

(2) Lines 55-60: The sentence is too long and hardly to understand the logic behind it.

**Reply:** Thanks for pointing out. We agree that this sentence was too convoluted, and we have rephrased it as follows.

*However, there is evidence that diabatic heating within storms tends to amplify their growth rates (e.g., Kuo et al., 1991; Davis et al., 1993; Stoelinga, 1996; Chang et al., 2002; Schemm et al., 2013). At the same time, transient diabatic processes, including surface fluxes, have been found to reduce eddy available potential (EAPE) and, thus, the reservoir from which eddy kinetic energy (EKE) can be generated (Ulbrich and Speth, 1991; Chang and Zurita-Gotor, 2007; Marcheggiani and Ambaum, 2020).*

(3) Line 84: I don't understand why you use "an otherwise" here, maybe delete that?

**Reply:** We agree that "otherwise" is unnecessary and we have removed the word.

(4) Line 90-92: I suggest the author to revise this sentence. Maybe from the perspective of EAPE/EKE budget analysis?

**Reply:** We have rephrased the sentence as:

*The response is described in terms of changes of the eddy energy cycle, that is EAPE and EKE, and the associated tendencies, including baroclinic, barotropic, and diabatic conversions, as well as the baroclinic conversion efficiency.*

(5) Line 121: remove "the" before "there is no seasonality…"

**Reply:** Done.

(6) Lines 123 and 124: the EKE represents eddy kinetic energy. Please add the word "energy" after "eddy kinetic" in the two places.

**Reply:** Thanks, we have corrected this.

(7) Figure 4: It is perhaps better to highlight the key region of conversion efficiency downstream of SST front in Figure 4. This is helpful for the reader to understand that the downstream enhancement of baroclinic conversion is caused by the conversion efficiency instead of baroclinicity or eddy total energy.

**Reply:** We agree and have marked the region with a star.

---

## Author Comment (AC2)

Reply to reviewer's comments #2

Summary: This paper examines the response of the storm track to uniform warming in an aquaplanet configuration of a GCM with a SST gradient added over a fixed longitudinal range in the midlatitudes. The paper also examines the response of individual cyclones. This paper is written in a clear manner, and the experimental design is interesting. The analysis utilizes a robust and useful EKE budget/framework that the authors have developed. The figures are all easy to interpret. The explanation for cause and effect are ok, but not expansive or 100% convincing, especially in terms of the regions that have a decrease in storm track activity within the tripole pattern.

My recommendation: minor but necessary revisions

**Reply**: We would like to thank the reviewer for this useful review of our study, which helped us improve the discussion of the EKE and EAPE response to warming. In response to the comments, we now address the role of stationary waves in the revised manuscript. As it turns out, flow modification by stationary waves contributes to local EKE and EAPE changes through changes in EKE and EAPE advection, especially at polar latitudes. However, the focus of the study remains on the role of EKE and EAPE sources and sinks. Below is our detailed response.

Minor comment that applies to the Abstract and the Conclusion sections.

(1) As I read the paper, I couldn't stop thinking about the stationary wave and its projected response to changes in the tropics with anthropogenic climate change (e.g., Wills et al. 2019). I note that this Wills paper does not provide a solution, and in fact does not touch directly on the issue of the storm tracks and stationary waves interacting (see the paragraphs under Fig. 4 in the Wills et al. paper). However, I think that this manuscript should include a bit more discussion on the role of the changing stationary waves in the CMIP6 models.

(2) In the introduction, the authors discuss previous work that shows a role for changes in the tropics and subtropics affecting the storm tracks. But these ideas are not revisited later in the conclusion section. And the abstract seems to suggest that the entire response can be captured in an atmosphere without stationary waves. For me, it is tricky to understand how their model captures the tripole pattern of the North Atlantic storm track response to anthropogenic warming in a model with no mountains. Perhaps I am just mis-reading the abstract and conclusions – i.e., the change found in their idealized model is suggestive of the tripole pattern but incomplete – in which case, simple changes to the wording and some additional caveats would suffice.

Or, if you feel that the storm track response in your idealized model matches well with the CMIP6 models, then I would appreciate if the authors add a discussion in the conclusion section explaining why the presence of a stationary wave is not necessary for capturing the North Atlantic storm track response.

**Reply**: Thanks for these comments. Since they are partially related to each other, we answer them together.

We agree that the projected response of stationary waves likely plays an important role in the storm track response. In this study, we would like to emphasize that the SST front alone can lead to such a tripole pattern – a circumstance that has not yet received much attention in the literature. Furthermore, we then provide a mechanistic explanation for this tripole pattern based on the principal source terms of EAPE and EKE, as well as the changes to stationary wave patterns (see below). However, we content that this does not imply, that in CMIP models the SST front is the sole cause of this tripole pattern. Specifically, changes of stationary waves or the North Atlantic warming hole might have an equally or even more important impact. We rephrased the abstract to make clear that an aqua-planet simulation cannot reproduce the full response. In the abstract we state "*(...) the tripole pattern is **qualitatively** reproduced by simulating the change (... )*".

Regarding the stationary wave response, it is also interesting to note that on the aquaplanet the stationary wave response to warming downstream of the SST front contributes to the generation of the tripole pattern. More specifically, the reduction in EKE and EAPE in "region 3" is partly due to the enhanced equatorward advection of EKE and EAPE on the western flank of the enhanced stationary trough downstream and poleward of the SST front. In the revised manuscript, we have included subsection 3.4 discussing the stationary wave response and its impact on the EKE and EAPE redistribution and included a brief discussion in the summary.

We also added one sentence to the abstract: *Amplified stationary waves affect EKE and EAPE advection, which contributes to the local EKE and EAPE minimum at polar latitudes.*

(3) It might also be helpful to show the change in the 250 hPa jet in your 4K run for comparison with that from the CMIP6 models, e.g., Harvey et al. 2020, has the jet and the storm track available for a comparison. In it, you can see an intensification of the upper-level jet on the equatorward side of the Gulf Stream region. This corresponds to region 1 in your Figure 2 I think.

Harvey, B. J., Cook, P., Shaffrey, L. C., & Schiemann, R. (2020). The response of the northern hemisphere storm tracks and jet streams to climate change in the CMIP3, CMIP5, and CMIP6 climate models. Journal of Geophysical Research: Atmospheres, 125, e2020JD032701. https://doi.org/

Wills, R. C. J., R. H. White, and X. J. Levine, 2019: Northern Hemisphere Stationary Waves in a Changing Climate. Curr. Clim. Change Rep., 5, 372-389, doi:10.1007/s40641-019-00147-6.

**Reply**: Thanks for this suggestion and the references. In the revised manuscript we include a plot of the changes of the 300 hPa wind speeds as a measure of jet stream changes (Fig. D1a). In the warmed simulation on the aquaplanet, the jet stream is generally shifted poleward. This shift occurs in both hemispheres and is independent of the presence of an SST front. The presence of the SST front introduces zonal asymmetries to this shift pattern with a stronger amplification of the jet downstream and poleward of the front. This amplification is aligned with the intensified storm track as evident from EKE changes. In addition, a reduction in jet stream intensity occurs in polar regions north of the SST front – roughly aligned with region 3 of the tripolar EKE change pattern.

With regard to the comparison with projected changes in CMIP6 models, we note that the intensification of the jet downstream of the SST front occurs on the equatorward flank of the

jet in CMIP6 models, whereas it occurs on the poleward flank on the aquaplanet. We can only speculate about the reasons for these discrepancies. On the aquaplanet, most of the changes is related to the eddy-mean-flow interactions, e.g., a strengthening of the storm track causing more eddy momentum flux convergence. This is supported by the fact that in the presence of the SST front, the zonal asymmetries in the jet are closely aligned with the changes in the storm tracks. In CMIP6 models, in contrast, other processes such as changes of the thermal driving of the jet as well as the topographically forced stationary waves (that are absent in our simulations) are very likely superposed on the eddy-driven change patterns.

In the revised manuscript, we have included references to Wills et al. (2019) and Harvey et al. (2020) in the last paragraph of section 3.1:

*The stronger signal over the North Atlantic suggests that likely other mechanisms are at play, including changes to stationary waves (Wills et al., 2019) and the jet stream (Harvey et al., 2020), as well as the formation of the North Atlantic warming hole.*

Line-by-line minor comments:

(4) L50: I would argue that Brayshaw et al. show the SST gradient to be secondary or tertiary, with the upstream mountain and the land-sea contrast both having more significant roles in setting the location and orientation of the Atlantic storm track.

**Reply**: Thanks for this comment. You are right about the secondary nature of the SST gradient alone. In line with your comment (6), we have rephrased the sentence as follows:

*The western boundary currents amplify the land-sea contrast and contribute to the anchoring of the oceanic storm tracks as they help maintain a near-surface zone of enhanced baroclinicity and supply heat and moisture from below by means of sensible and latent heat fluxes (Chang et al., 2002; Sampe et al., 2010; Brayshaw et al., 2011; Papritz and Spengler, 2015).*

(5) L53-60: I agree with this summary. I think there are a few other recent papers that you might take a look at as they provide additional context for thinking about the role of SST fronts and storm intensification, e.g.,: Tsoporidis et al. 2021 and/or: Reeder et al. 2021

Tsopouridis, L., Spengler, T., and Spensberger, C.: Smoother versus sharper Gulf Stream and Kuroshio sea surface temperature fronts: effects on cyclones and climatology, Weather Clim. Dynam., 2, 953–970, https://doi.org/10.5194/wcd-2-953-2021, 2021.

Reeder, M. J., Spengler, T., & Spensberger, C. (2021). The Effect of Sea Surface Temperature Fronts on Atmospheric Frontogenesis, Journal of the Atmospheric Sciences, 78(6), 1753-1771.

**Reply**: Thanks for pointing us towards these references which are indeed highly relevant here. We have added the following sentences:

*These reinforcements occur mainly between rather than during the passage of cyclones and fronts (Marcheggiani and Ambaum 2020, Reeder et al., 2021; Tsopouridis et al., 2021). Thus, the SST front acts to pre-condition the environment for subsequent cyclones.*

(6) L111: Given the strength and spatial extent of the SST gradient that you are imposing, its spatial scale, and the fact that you have tilted it off the zonal access, it seems like what you are

doing is replicating a combination of the Gulf Stream and the land-sea contrast that exists in winter in the vicinity of the Gulf Stream extension. I think it would be best to include some statement to this effect.

**Reply**: Yes, we agree that by imposing rotated positive and negative anomalies we mimic the behavior of the Gulf Stream and the land-sea contrast. Note that imposing an SST dipole instead of just a positive anomaly is necessary to avoid a net warming effect due to the presence of the front and to allow for a meaningful comparison between the northern and southern hemisphere (i.e., hemisphere with and without SST fronts). We have added the following sentence to the methods section:

*By imposing rotated positive and negative SST anomalies we mimic the combined effect of the Gulf Stream and the land-sea contrast in the western North Atlantic.*

(7) Related to my big-picture comment above: the manner in which you have added the SST gradient implies that you are including something of a stationary wave in the model. I think this is something work mentioning. In your discussion of Fig 1a, you refer to a trough in the EKE field but I am curious to see the Z500 anomaly with respect to the zonal mean. Also, I wonder how the storm track responds of the SST front is not tilted, which is more like the Kuroshio.

**Reply**: The presence of the SST front does lead to a stationary wave response, comprising a trough immediately downstream and poleward of the front as well as a transition to a ridge about 90° eastward of the front. This stationary wave response, ultimately resulting from eddy-mean flow feedback, was previously shown by Kaspi and Schneider (2011) to limit the downstream extent of a storm track. In the warmed simulation, this stationary wave response is amplified. In the revised manuscript we include Fig. D1b showing the change of the departure of 500 hPa geopotential from the zonal mean.

Investigating the influence of the tilt of the SST front on the warming response and, in particular, whether it might explain different responses of the North Atlantic and the North Pacific storm tracks to warming is an interesting question, which we would like to reserve for future work. We only note that control simulations with no tilt in the SST front showed qualitatively similar characteristics of the storm track. However, we did not yet perform simulations with warming.

(8) L213: The difference in the Pacific is likely also related to the lack of co-location of the western boundary current and the coastline, and the fact that the Gulf stream has more north/south variation whereas the Kuroshio is relatively zonal and, there are clear differences in the meridional orientations of the jets above the western boundary currents in the Pacific (more zonal jet) and the Atlantic (meridionally tilted jet due to many factors, see L50 comment).

**Reply**: We agree, the more zonal orientation of the SST front and the jet stream likely contribute too. Thanks for suggesting. We have rephrased the sentence as follows:

*In contrast to the Gulf Stream SST front, the EKE change over the Kuroshio is less pronounced (Figs. 2a,b and c). Possibly, this is related to climatologically weaker SST gradients over the western North Pacific, a more zonal orientation of the SST front and the jet stream, as well as*

*the distance of the SST front from the continent. Nevertheless, we note that the change pattern is also not absent.*

(9) L277-279, you write:

"Apparently to the north of the SST front and downstream the re-organization by the SST front in the warmer climate is such that external baroclinic conversion efficiency increases, …"

This sentence is hard for me to follow. When I read it, I assume it is missing a comma after the word downstream. Is that all, or it there also a word missing somewhere?

**Reply**: Yes, there should be a comma after downstream. We have rephrased the sentence for better clarity as follows:

*Hence, we conclude that as the external baroclinic conversion does not change globally, the SST front in the first place acts to localize the external baroclinic conversion downstream of the front. This localization is such that in the warmer climate, external baroclinic conversion efficiency increases northeast of the SST front. This indicates a better alignment of the eddy heat flux with the mean baroclinicity compared to the control. As a consequence, the eddies become more efficient in tapping into the mean potential energy reservoir.*

(10) L280: Your interpretation of the intensification being related to the eddies becoming more efficient in tapping the mean potential energy reservoir is interesting to me. By this same logic, in the region where baroclinic conversion of EAPE decreases, especially in the area south of the SST front, is your interpretation that the eddies in this region have become less efficient? Or is it simply that there are less eddies in that region?

**Reply**: Reduced baroclinic conversion in the area south(-west) of the SST front, i.e., region 1, appears to result from a combination of a reduced eddy efficiency and the poleward shift of background baroclinicity. We have rephrased the corresponding paragraph slightly to make more clear how these changes affect the tripole pattern.

(11) L 297-300, you write:

"The change in resolved-scale condensation and evaporation, i.e., due to saturation adjustment, is of similar importance near the SST front in terms of magnitude. However, the absence of a clear dipole structure in the zonal mean indicates that the global increase in resolved-scale condensation and evaporation is stronger compared to that resulting from the cloud microphysics parameterization (Fig. 5b)."

I am not sure what you mean by this last sentence. The lack of a dipole in the zonal mean indicates what? And isn't the lack of a dipole highlighting the fact that the condensation (presumably in the storms' warm sectors) is the process that is changing the most with warming?

How does the lack of a dipole imply something about the cloud microphysics? I see that cloud microphysics does have a dipole, so I am have an idea where you are going with this, but could you expand on this?

Reply: We agree that this sentence is not totally clear. The lack of a dipole in the zonal mean indicates that there is a net increase, while the presence of a dipole of similar amplitude indicates that cloud microphysics are reduced at low latitudes but increased at high latitudes by the same amount. Hence, they are merely shifted poleward with the storm track without a global increase. We re-phrased the sentence:

*However, the zonal mean indicates that much of the increase is globally compensated by a decrease further equatorward. In contrast, the change in resolved-scale condensation and evaporation, i.e., due to saturation adjustment, is of similar importance near the SST front in terms of magnitude and the absence of a clear dipole structure in the zonal mean indicates no compensation elsewhere.*

(12) L359: Figure 7: I would find it helpful if you placed the numbers 1,2,3 from Figure 2d on Figure 7. My sense is that the location of the changes in cyclone track density are not 1-to-1 with the location of the change in EKE. This is not a huge surprise. But some discussion of this would be nice.

**Reply**: We understand that it is tempting to overlay the numbers on the different figures, and also added the labels to Fig. 7, but as you correctly anticipate they are not co-located with regions of cyclone anomalies. Previous studies have shown that feature-based surface storm tracks are located far poleward of regions highlighted by EKE. The former is frequent where cyclones are mature and propagation is slow, which is where EKE is low. A detailed discussion of the relationship between the two perspectives can be found in Fig. 1 in Schemm and Schneider (2018).

Schemm, Sebastian, and Tapio Schneider. " Eddy Lifetime, Number, and Diffusivity and the Suppression of Eddy Kinetic Energy in Midwinter". Journal of Climate 31.14 (2018): 5649-5665. https://doi.org/10.1175/JCLI-D-17-0644.1.

(13) L408-409, you write:

"the interaction between the diabatically-induced circulation and the cyclone also advects the cyclone poleward."

Is this statement shown already in a specific paper? It seems like this would be another one that depends critically on the location the strongest diabatically-induced circulation.

**Reply**: Yes, there are several previous studies that analyze this relationship and which we added to this statement.

Coronel, B., Ricard, D., Rivière, G., and Arbogast, P.: Role of moist processes in the tracks of idealized midlatitude surface cyclones, J. Atmos. Sci., 72, 2979–2996, https://doi.org/10.1175/JAS-D-14-0337.1, 2015.

Gilet, J.-B., Plu, M., and Rivière, G.: Nonlinear baroclinic dynamics of surface cyclones crossing a zonal jet, J. Atmos. Sci., 66, 3021–3041, https://doi.org/10.1175/2009JAS3086.1, 2009.

Rivière, G., Arbogast, P., Lapeyre, G., and Maynard, K.: A potential vorticity perspective on the motion of a mid-latitude winter storm, Geophys. Res. Lett., 39, L12808, https://doi.org/10.1029/2012GL052440, 2012

Tamarin, T. and Kaspi, Y.: The poleward motion of extratropical cyclones from a potential vorticity tendency analysis, J. Atmos. Sci., 73, 1687–1707, https://doi.org/10.1175/JAS-D-15-0168.1, 2016.

(14) L411: Do the poleward displacement changes have any bearing on the SST tripole in the storm tracks?

Related to this minor comment, here is a commentary that the authors can take action on or not, I leave it to you: the Lagrangian tracks element and the storm track + EKE element are a bit disconnected. Both relate to the SST front, but otherwise, the two elements are not currently woven together into a single story. This is not a game changer, as it stands, it seems a bit like you have two separate components in this manuscript.

**Reply:** Yes, we agree with your comment. It is challenging to combine the two historically divided perspectives into a coherent story. There is no solution to the "problem" that both cannot easily be woven into a single story. In the past, we circumvented this and tracked EKE and EAPE source terms along the cyclone tracks (e.g., Schemm & Schneider 2018, Schemm & Rivière 2019) but this approach comes with its own deficits. The amount of work would produce sufficient material for a full second study. The idea behind the surface cyclone track frequency (Fig. 7) is to build a bridge between the two perspectives.

Schemm, Sebastian, and Tapio Schneider. " Eddy Lifetime, Number, and Diffusivity and the Suppression of Eddy Kinetic Energy in Midwinter". Journal of Climate 31.14 (2018): 5649-5665. https://doi.org/10.1175/JCLI-D-17-0644.1

Schemm, Sebastian, and Gwendal Rivière. " On the Efficiency of Baroclinic Eddy Growth and How It Reduces the North Pacific Storm-Track Intensity in Midwinter". *Journal of Climate* 32.23 (2019): 8373-8398. < https://doi.org/10.1175/JCLI-D-19-0115.1>.

The poleward displacement has been linked to the general poleward shift of the storm track (Tamarin and Kaspi 2017) but we can only speculate as to whether it affects the tripole pattern exactly. Likely, cyclones downstream of the SST front deepen and propagate more systematically within similar regions and latitude bands. As their deepening rates are enhanced, we would expect a local EKE increase regionally aligned with the stronger poleward displacement.

(15) L435: You write:

"The SST front organizes the flow such that the baroclinic growth becomes more efficient downstream of the front, that is, the eddy heat flux better aligns with the baroclinicity vector."

This to me, is the one statement in the paper (and I acknowledge that this is also stated in the results section) where the author offer an answer as to why the present of the SST front might lead to the tripole pattern in the storm track response. Maybe I am missed some other statements on the matter? As it stands, this statement offers an explanation for the increase in the storm track, but it does not give an explanation as to why there are the two minima. Would any

disruption in the zonal flow lead to this same response? You could at least test the question of the role of the tilt in the SST, at that would have some correspondence to the Pacific vs the Atlantic storm track basin.

**Reply**: We agree with you that our focus in this section is strongly on the EAPE and EKE increase ("region 2"). In the revised version more attention is given to the two local minima. Reduction in EKE and EAPE in the polar "region 3" is partly due to the enhanced equatorward advection of EKE and EAPE on the western flank of the enhanced stationary trough downstream and poleward of the SST front. We added corresponding statement into our manuscript. The reduction in the upstream "region 1" is due to a reduced efficiency (Fig. 4), which in this region is stronger than at similar longitudes elsewhere. We also added a statement into the corresponding section 3.2 and into the summary. The strong SST front in our simulation, which mimics the Gulf Stream plus the land-sea contrast, is an effective disruption of the zonal flow and we would expect comparable tripolar shift pattern of the jet for the case of a mountain but with differing changes in EKE/EAPE amplitude.

---

## Author Response (AR2)

Reply to reviewer's comments #1 to revised manuscript

Summary

I thank the authors for considering all of my prior comments and making updates. I think the manuscript reads well. My only remaining comment is a request. In the limitation section (5.1), I think it would be useful to mention that the response of the storm track in your integrations are weaker than those found in the models, and add some comment about the likelihood that a reason for this is the lack of a land-sea contrast and upstream mountain range. This idea links back to the Brayshaw et al work and the fact that the ocean SST gradient is but one of many important factors for the storm track in the North Atlantic. The reason I am so particular about this is because I have long dealt with some scientists who insist on the outsized role of the Gulf Stream. As this manuscript currently stands, they will read it and see no subtly. But if you have these caveats here, then perhaps that response will be somewhat different.

**Reply:** We agree with your comment, and it is useful to highlight in the limitation section that the strength and orientation of the negative and positive SST anomaly mimics the combined effect of land-sea contrast and Gulf Stream front and not only the latter. Now, this is mentioned only in the data & methods section. Our setup is not considering the influence of an upstream mountain range (i.e., Rocky Mountains) and an interactive ocean. The latter is known to damp the storm track response, which could explain why the EKE response in our simulation is twice as large compared to that seen in CMIP models (Fig. 2). We added the following remark to the limitation section:

*It should be noted that the zonal asymmetry created by the rotated SST anomalies unlikely reflects the influence of the Gulf Stream front alone but rather the combined land-sea contrast including the Gulf Stream SST front. The lack of an upstream mountain range and an interactive ocean could be one reason for the different magnitude of EKE change in our idealized model compared to the CMIP models.*